# How Does Critical Batch Size Scale in Pre-training?

**Hanlin Zhang**[1]    **Depen Morwani**[1,2]    **Nikhil Vyas**[1]    **Jingfeng Wu**[3]
**Difan Zou**[4]    **Udaya Ghai**[5]    **Dean Foster**[5]    **Sham Kakade**[1,2]
[1]Harvard University    [2]Kempner Institute, Harvard University
[3]University of California, Berkeley    [4]The University of Hong Kong    [5]Amazon

## Abstract

Training large-scale models under given resources requires careful design of parallelism strategies. In particular, the efficiency notion of critical batch size (CBS), concerning the compromise between time and compute, marks the threshold beyond which greater data parallelism leads to diminishing returns. To operationalize it, we propose a measure of CBS and pre-train a series of auto-regressive language models, ranging from 85 million to 1.2 billion parameters, on the C4 dataset. Through extensive hyper-parameter sweeps and careful control of factors such as batch size, momentum, and learning rate along with its scheduling, we systematically investigate the impact of scale on CBS. Then we fit scaling laws with respect to model and data sizes to decouple their effects. Overall, our results demonstrate that CBS scales primarily with data size rather than model size, a finding we justify theoretically through the analysis of infinite-width limits of neural networks and infinite-dimensional least squares regression. Of independent interest, we highlight the importance of common hyper-parameter choices and strategies for studying large-scale pre-training beyond fixed training durations.[1]

## 1 Introduction

Efficient optimization is critical in pre-training large models (LMs) at scale (McCandlish et al., 2018; Shoeybi et al., 2019; Kaplan et al., 2020). In particular, large-batch training is key to accelerating training, as it enables more efficient parallelism across hardware accelerators (You et al., 2017; Goyal et al., 2018). Specifically, understanding the scaling behavior of the *critical batch size* (CBS) is essential for optimizing data parallelism, as it defines the point beyond which increasing the batch size may result in computational efficiency degradation. Below the CBS, approximately *linear scaling* is achievable—doubling the batch size can proportionally reduce the number of optimization steps required to reach a target loss. However, beyond this threshold, further increases in batch size would lead to diminishing returns, making it essential to balance computational efficiency with model performance (Shallue et al., 2019; McCandlish et al., 2018). This trade-off presents a challenge for studying pre-training given resource constraints as practitioners are compelled to navigate difficult decisions in balancing compute, data, and training time.

We investigate the scaling laws governing CBS in the context of autoregressive transformer-based language modeling (Vaswani, 2017; Radford et al., 2018). Analyzing CBS in pre-training is challenging due to the absence of a precise formalism relating it to model and data sizes in the literature (Kaplan et al., 2020; McCandlish et al., 2018). Moreover, the interwined effects of scaling model and data sizes proportionally (Hoffmann et al., 2022b) further complicate this analysis. Although previous works study the effects of batch size on optimization performance (DeepSeek-AI et al., 2024; Besiroglu et al., 2024; Porian et al., 2024), two crucial differences are (1) they do not decouple model size and data size; (2) they focus on optimal batch size that reaches the minimum loss instead of critical batch size. We measure critical batch size as a metric, which represents the batch size that results in certain overhead compared to linear scaling: given a certain target validation loss,

---

[1]Code available at https://github.com/hlzhang109/critical-batch-size.

**Empirical Takeaways :**

1. In Chinchilla settings, CBS increases when model size $N$ and data size $D$ (or training duration thereafter, which we will use interchangeably) are jointly scaled up (Figure 1, left).

2. If we scale up training duration $D$ while keeping $N$ fixed (Figure 1, middle), the critical batch size increases to a similar degree.

3. However, we find that CBS remains nearly invariant when scaling up $N$ while keeping $D$ fixed (Figure 1, right), suggesting that CBS weakly depends on model size $N$ but more strongly depends on data size $D$.

4. Our experiments on small 151M proxy models provide insights into a range of common hyper-parameters and optimization configurations, including transformer context length adjustments, and scaling strategies based on width versus depth, among others.

we measure the number of steps to reach it for different batch sizes; we derive the transition points that incur certain overhead when doubling the batch size; then we fit scaling laws *w.r.t.* model size and data size to systematically study the scaling of CBS.

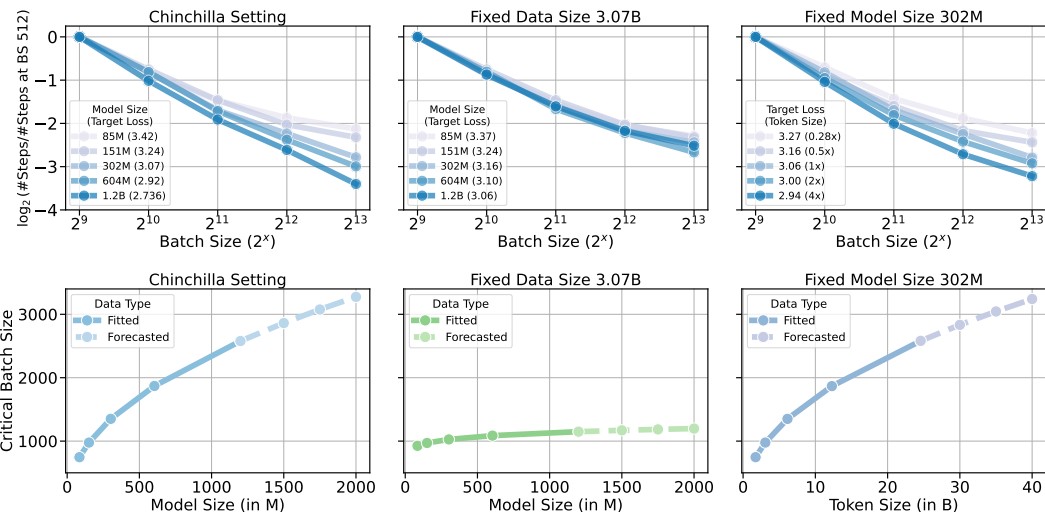

Figure 1: **Optimization efficiency and scaling of critical batch size in Chinchilla (left) and controlled (middle, right) settings**. To study the effect of CBS across different model sizes, we track the relative number of steps required to reach a certain target validation loss. In the Chinchilla setting (left), we keep the data-to-model size ratio $D/N = C_{\text{Chin}}$ constant and observe that CBS increases with scale. However, when controlling for either **model size (middle)** or **data size (right)**, the growth in target losses becomes mostly dependent on data size rather than model size (Section 3).

## 1.1 EMPIRICAL TAKEAWAYS

Conceptually, we formalize the notion of critical batch size and examine the independent effects of both model and data size. We start by scaling up data size in tandem with model size, as suggested in the Chinchilla compute-optimal framework (Hoffmann et al., 2022b). Through controlled studies, we propose scaling laws that decouple the growth of critical batch size from model and data size, leading to the following takeaways — an aspect underexplored in previous research.

Overall, our empirical finding that **CBS scales primarily with data size** implies that when scaling up data, one can reduce serial training time through greater data parallelism due to the increase of CBS, without a loss in computational efficiency that can be measured by floating point operations (FLOPs).

## 1.2 THEORETICAL IMPLICATIONS

Theoretically, maximal update parameterization suggests that, beyond a certain point, increasing the width of the neural network (while keeping data size fixed) does not further increase the critical batch size. In contrast, by analyzing a simple least-squares regression with mini-batch SGD, we provide a theoretical basis for how the critical batch size continues to scale with increasing data size. Specifically, we introduce two informal theorems here and refer readers to Section 4 for more details.

**Theorem 1** (Informal version of Theorem 2). *In infinite width regimes (Yang & Hu, 2021), training dynamics and performance of the networks become effectively independent of the model size. Consequently, the critical batch size remains nearly invariant when scaling up the model size beyond this point, indicating that larger models do not require proportionally larger batch sizes to achieve optimal training efficiency.*

**Corollary 1** (Informal version of Corollary 2). *Consider mini-batch SGD with $D$ samples in the least square problems under power-law source and capacity conditions. The CBS, which enables mini-batch SGD to achieve the minimal expected excess risk while ensuring the fastest possible serial runtime, is given by $B^*(D) = \Theta(D^c)$, where the exponent $c \geq 0$ is determined by the exponents of the source and capacity conditions. In the regime where the variance error tends to be dominant, we have $0 < c < 1/2$, indicating CBS grows with data size.*

## 2 EXPERIMENTAL DESIGN AND EMPIRICAL FINDINGS

We describe the experimental settings and refer readers to Appendix D for extra details. Throughout this paper, we use the abbreviations 'M' for million, 'B' for billion, and 'T' for trillion.

### 2.1 EXPERIMENTAL SETTINGS

**Model and training details**. We train a series of autoregressive LMs with a context length of 512 in different sizes ranging from 85M, 151M, 302M, 604M to 1.2B (Appendix D Table 2) on C4 (Raffel et al., 2020) using Adam (Kingma, 2014) with optimizer-specific hyper-parameters reported in Table 3. We adopt the tokenizer of Eleuther AI's gpt-neox-20b that has a vocabulary of size 50280. We use small 151M proxy models to analyze hyper-parameters in most ablation studies. We set the micro batch size smaller than the global one and use gradient accumulation to simulate the effects of large global batch sizes. We focus on fully synchronized distributed data parallelism scenarios where communication is frequent, which simplifies the evaluation and abstracts actual wall clock savings into a total number of optimization steps. More details on optimizer configurations and evaluation strategies are included in Appendix D.

**Experimental design and outline**. To study CBS in Chinchilla settings, we need to consider target loss on a holdout validation set and measure the number of optimization steps required to reach it. We consider the optimal batch size as $B_{\text{opt}}$ in the linear scaling regime that incurs no efficiency overhead as detailed in Appendix C. We consider the validation loss of an optimal batch size $B_{\text{opt}} = 256$ at step $t_{\text{Chin}} = C_{\text{Chin}} \times N/(\text{ctx\_len} \times B)$ for each model size $N$ and batch size $B$ with context length ctx\_len set to be 512, where $C_{\text{Chin}}$ is the Chinchilla coefficient.

When scaling up model size jointly with data size, the above implies that each model size would have a different target loss. Achieving such a goal through the procedure above is challenging not only due to the combinatorially many hyper-parameters but also the unknown training dynamics of each model size and batch size. Below, we outline several key aspects to approach the goal:

1. As we focus on the number of training steps needed to achieve a target validation loss, learning rate decay strategies typically require predefining the total training duration (Loshchilov & Hutter, 2022; Hu et al., 2024; Hägele et al., 2024; Defazio et al., 2024). To address this, we propose using *exponential weight averaging* (EWA) (Polyak & Juditsky, 1992) to achieve the desired target validation loss, a simple approach that matches other popular choices (Figure 2). This enables training beyond fixed durations or data size, allowing to resume training from checkpoints until the target validation loss is achieved (Section 2.2).

2. Training with proper hyper-parameters: ensuring proper sweeps of momentum and learning rate (Appendix A); adopting well-tuned values for the $\beta_2$ parameter and the exponential weight averaging decay rate $\tau$, tailored to each batch size (Appendix B).

## 2.2 TRAINING BEYOND FIXED DURATIONS FOR REACHING TARGET VALIDATION LOSS

**Benchmarking learning rate schedulers.** In practice, LMs are usually trained using fixed token budgets (Hoffmann et al., 2022b), which can determine the total number of iterations the training would undergo. This training process can be easily decomposed into learning rate warmup and decay phases so that a lower learning rate is kept at the end of training to enable better optimization. However, our goal is to find the optimal performing run under various hyper-parameters and optimization conditions. This implies a non-trivial decision regarding selecting the maximum training duration (Defazio et al., 2024; Hägele et al., 2024). As training beyond fixed durations is particularly favorable in many large-scale pre-training scenarios, we benchmark recently proposed methods like schedule-free optimizer (Defazio et al., 2024), cosine, warmup-stable-decay schedule (WSD) (Hu et al., 2024) (or trapezoidal (Zhai et al., 2022)) and our proposed constant+EWA strat-

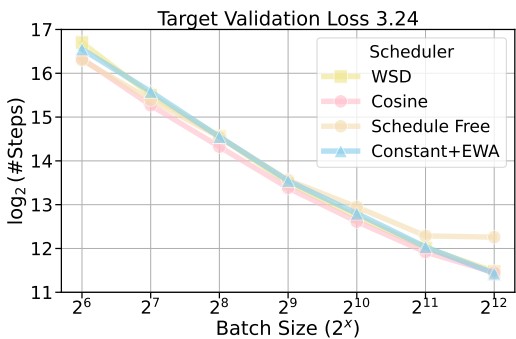

Figure 2: **Comparing and accounting for training dynamics.** Throughout, we adopt **Constant+EWA** since it performs the best for large batch sizes and avoids setting a fixed training duration beforehand for reaching a target loss.

egy which maintains a running average of model weights $\xi_{t+1} = \tau \cdot \xi_t + (1 - \tau) \cdot \theta_t$ to improve optimization, where $\theta_t$ is the actual model parameter at step $t$ and we use $\xi$ for evaluation. To ensure the baselines are close to optimal, we first get the optimal step counts from our constant+EWA runs, we evaluate WSD scheduling by testing [0.1, 0.2, 0.3] multiples of those step counts. In parallel, for the cosine scheduler, we explore [0.9, 1.0, 1.1] times the same step counts. We show that our constant+EWA strategy can match the efficiency of cosine scheduling and WSD, especially for large batch sizes (Figure 2). The connections are explained in detail in (Morwani et al., 2025).

Prior research has shown the generalization (Izmailov et al., 2018) and optimization (Karras et al., 2024) benefits of EWA, our findings further reveal that EWA can trade memory for optimization efficiency in LM pre-training, especially in large-batch regimes. This is useful in scenarios where a target loss must be achieved, but practitioners are uncertain of the exact maximum data size to set up the learning rate schedule for training.

> **Takeaway on learning rate scheduling:** EWA consistently improves model training efficiency compared to using a constant learning rate without it. EWA proves to be an effective approach compared to other baselines with decaying schemes, offering competitive performance while eliminating the need to predefine training durations.

## 2.3 ABLATION ON MODEL CONTEXT LENGTH

We adopt 512 as the context length of LMs for all of our experiments but it is unclear how it would impact the efficiency of training and whether the scaling of CBS would vary when we enlarge the context length. So we sweep over several larger windows $2^{10}, 2^{11}, 2^{12}$ (Section 2.4) Overall all models in four different context lengths have very similar relative optimization efficiency across various batch sizes and thus justifies our use of 512 for all the experiments.

> **Takeaway on model context length:** Different context lengths ($2^9 \sim 2^{12}$) have similar scaling *w.r.t.* batch size.

## 2.4 ABLATION ON MODEL WIDTH AND DEPTH

Model sizes can typically be scaled up in two main ways: by increasing the width, which involves enlarging the hidden size of the multilayer perceptron (MLP), or by increasing the depth, which entails adding more layers to the network. As the main result in Figure 1 only involves a single way for scaling up models (Table 2), e.g. 604M model has $2\times$ width than the 151M one. To explore alternative scaling strategies, we investigate how the model behavior changes when we scale the 604M model by increasing the depth by $4\times$ instead (detailed configurations in Table 5).

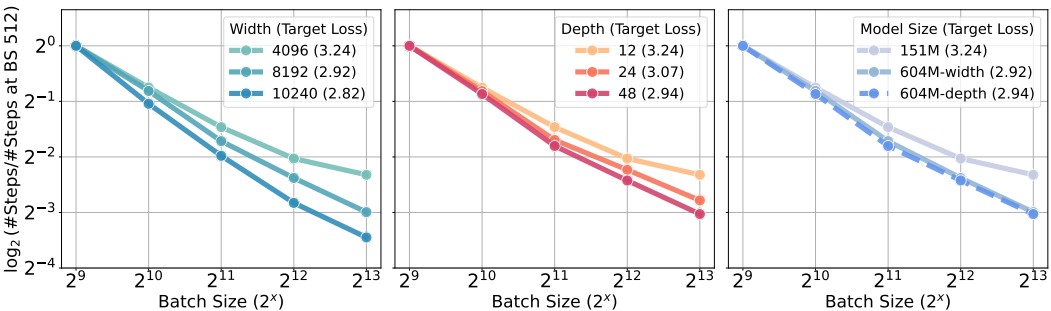

Figure 3: Scaling up width and depth shares similar efficiency gain for compute-optimal training.

Firstly, as shown in Figure 3 (left, middle), under Chinchilla scaling—where data and model size grow **proportionally**— increasing model depth or width has a similar impact on CBS. Notably, according to previous results, the rise in CBS in compute-optimal scaling should be attributed to the increased data size. Then through controlled comparison (Figure 3, right), we see that using two different ways to scale 151M models to 604M ones is equivalent in efficiency since both curves overlap. Our findings may offer practical insights for scaling models under a fixed token budget that is allocated in proportion to model size. This is particularly relevant because scaling model width is often favored over increasing depth, as wider models tend to be more amenable to parallelization without incurring additional latency overhead (Shoeybi et al., 2019; Touvron et al., 2023; Erdil, 2024; McLeish et al., 2025).

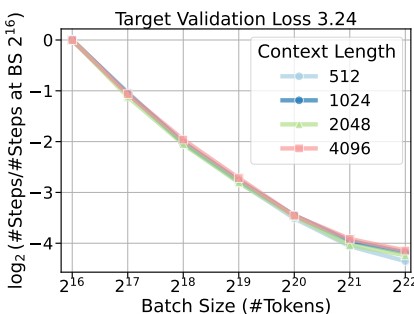

Figure 4: Ablation results on **context length** using 151M models.

> **Takeaway on scaling transformer width and depth in compute-optimal regimes:** Increasing width and depth has similar effects in critical batch size for compute-optimal pre-training.

## 3 CRITICAL BATCH SIZE SCALING LAW

### 3.1 FORMAL DEFINITION OF CRITICAL BATCH SIZE

Recall that CBS is the transition point where increasing the batch size by a factor of $k$, leads to a reduction in the required number of training steps by a factor that is less than $k$. We now define CBS as the batch size that leads to a $20\%$ overhead compared to linear scaling. First of all, define $\mathcal{R}(N, D, B)$ as the best loss achievable for a model of size $N$ using a single pass on $D$ tokens with a batch size $B$. This would be obtained by optimally tuning all other parameters of the optimizer, while keeping $N, D, B$ fixed. Below is the formal definition of CBS:

**Definition 1.** *Define $\mathcal{R}_{opt}(N, D) = \min_B$ $\mathcal{R}(N, D, B), B_{opt}(N, D) = \arg\min_B \mathcal{R}(N, D, B)$, as the minimal loss achieved optimizing over batch size and the optimal batch size respectively. We define $f_{N,D}(B)$ to be the number of steps required to reach $\mathcal{R}_{opt}(N, D)$ as a function of batch size $B$. Clearly $f_{N,D}(B_{opt}) = D/B_{opt}$. To*

*define the Critical Batch Size, $B^*(N, D)$, we can define a linear scaling curve $f^*_{N,D}(B) = D/B$. $f^*$ matches $f$ at $B_{opt}$ and then scales down linearly as batch size goes up. $B^*(N, D)$ is defined as the maximum batch size $B' > B_{opt}(N, D)$ such that $f_{N,D}(B') \leq 1.2f^*_{N,D}(B')$.*

As illustrated in Figure 5, $B^*(N, D)$ is the batch size at which the number of steps is 20% higher than what is predicted by the linear extrapolation from the optimal batch size. Note here that 20% can be replaced by any other suitable measure of increase from linear scaling.

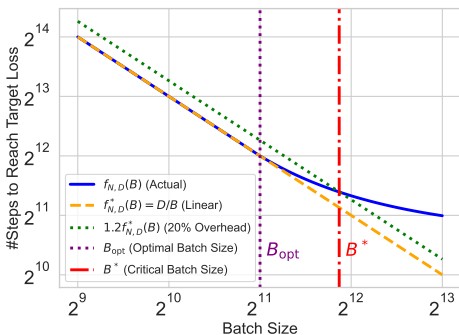

Figure 5: Illustration of critical batch size, where $B^* = 2^{11.87}$ and context length is 512 by default.

## 3.2 SCALING LAWS *w.r.t.* MODEL SIZE FOR CHINCHILLA-OPTIMAL PRE-TRAINING

As observed in all the results above, doubling the batch size for larger models allows them to more efficiently reduce the relative number of steps needed to reach the target loss. We ask whether those increased efficiencies can be predicted via a scaling law.

We begin our first step by fitting a power law of batch size ($B$) to the **absolute** number of steps ($Y$) to reach the target loss $\log(Y) = \log(a + \frac{b}{B^\alpha})$ and then derive the critical batch size. Then we derive the CBS via $B^* = \frac{b}{5a} + 1.2B_{opt}$, which is implied by a transition point where the total amount of data under this batch size would incur 20% overhead compared to linear scaling: $D_{total} = (a + b/B^\alpha_{opt}) * 1.2B_{opt} = (a + b/B^\alpha) * B$, where $\alpha = 1$, $B_{opt}$ is set to be 256 chosen to lie within the linear scaling regime as suggested in Appendix C. We report the parameters fitted to the power law relationship between the number of steps $Y$ and batch size $B$ in Appendix Table 6. We adopt the fixed $\alpha = 1$ solution, as both strategies yield nearly identical forecasting results.

Secondly, we fit a power law $\log(B^*) = \log(c + \frac{d}{N^\beta})$ with respect to the model size $N$ (in million). The constant term $c$ is set to be 0 by default (as $B^*$ should be 0 at $N = 0$), which leads to $B^* = 93.20 * N^{0.47}$. We visualize the curve fitted in Figure 1 (left) and report more forecasts in Table 7 in the Appendix.

Overall, we observe an increase in CBS when scaling up in compute-optimal training: In Figure 1 (left), we have target losses selected according to chinchilla steps and we fit the power law of critical batch size with respect to model sizes $N$ (in million) as $B^* = 93.20 * N^{0.47}$. Our results suggest that a critical batch size around $2^9$ to $2^{11}$ would be helpful to efficiently optimize models below 1B on Chinchilla-optimal amount of tokens to study other empirical problems. However, it is common for the number of tokens trained to scale proportionally with the model's parameter count. So it is unclear whether the growth of CBS is because of the increase in (1) model size or (2) the data size/training duration, a question we explore in the next subsection.

## 3.3 DECOUPLING CBS SCALING LAWS *w.r.t.* DATA SIZE AND MODEL SIZE

**Controlled comparison with the same data size.** Firstly, we use the Chinchilla token size 3.072B of 151M models $t_{Chin}$ to record the target validation loss for each model size and train all the 302M, 604M, 1.2B models with a smaller duration again to reach these target losses. To optimize for performance when training on fewer tokens, we also tune the warmup steps accordingly. Figure 1 (top right) shows that all the curves behave similarly and we observe almost no increase in CBS when enlarging the model size. Moreover, we fit a scaling law with respect to model size thereafter Figure 1 (bottom right): keeping the data size fixed leads to a scaling law $B^* = 621.341 * N^{0.087}$ weakly dependent on model size.

**Controlled comparison with the same model size.** Moreover, focusing on the 302M models, we conduct additional experiments by selecting target losses at $0.28\times$, $0.5\times$, $2\times$, and $4\times$ the Chinchilla step for batch size 256 runs. This setup results in two under-training and two over-training configurations. To achieve optimal performance in the over-training scenarios, we increase the warm-up

ratio accordingly, while for the under-training cases, we reduce the warm-up ratio proportionally. Results in Figure 1 (middle) show that as we enlarge the number of tokens being trained on, we see an increase of CBS, similar to what we have observed for training large models on chinchilla target loss. This can also be seen in the forecasted CBS curves shown in Figure 1 (middle) which shows that as we enlarge the number of tokens being trained on, we see an increase of CBS, similar to what we have observed in the Chinchilla setting where model and data size are scaled up proportionally.

We also plot the results for scaling both $N$ and $D$ (Figure 1, left) and only scaling $D$ (Figure 1, right) together in Figure 6. In the side-by-side comparison, we observe the following trends: (i) In the Chinchilla setting (indicated by the first column in the legend), models of various sizes (85M, 151M, 604M, 1.2B) trained on different token amounts exhibit an increase in critical batch size as scale grows. (ii) Additionally, each pair of curves with the same color overlaps significantly, indicating that models of different sizes trained on the same token quantity tend to have similar critical batch sizes. (iii) Finally, when model size is held constant and only the data size (second column in the legend) varies, we also observe an increase in critical batch size with scale. Therefore, we can qualitatively understand that the increase of CBS is likely to be agnostic to model sizes but due to the increase in training duration.

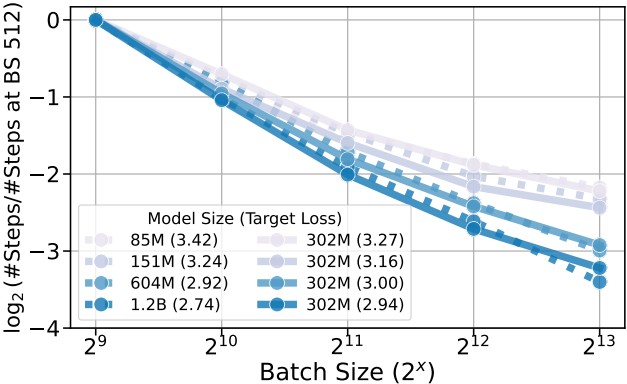

Figure 6: Controlled comparison by training **302M** models on varying amounts of tokens and then comparing with other model sizes trained in similar amounts of tokens. Models with the same color or positioned in the same row of the legend represent this comparison. For a fixed token count of 3.072B, we measure the target loss at that step for each model size.

> **Takeaway on scaling laws for critical batch sizes:** Based on the scaling laws and controlled comparisons, we conclude that the increase in CBS in Chinchilla-optimal training is more strongly attributed to extended data size or training durations rather than the increase of model size.

## 4 THEORY ON SCALING OF CRITICAL BATCH SIZE

Our experimental results show that CBS increases with larger data sizes but remains (nearly) invariant when scaling up the model size. We now formally investigate this observation using theoretical analysis for both scenarios.

### 4.1 FIXED DATA SIZE AND SCALING UP MODEL SIZE

Various previous works have established infinite width limits of neural networks (Yang & Hu, 2021; Bordelon & Pehlevan, 2022). For initializations and architectures that obey these limits, we can theoretically claim, that for a fixed training duration and batch size, the performance of the neural networks asymptotes with increasing width. The formal statement is provided below:

**Theorem 2.** *For SGD with a given batch size $B$ (or for gradient descent, i.e., $B \to \infty$), training iterations $t$, an error tolerance $\epsilon > 0$, fixed learning rate schedule and data ordering, for any network and initialization satisfying Master Theorem (Theorem G.4) in Yang & Hu (2021), there*

*exists a width $w$ such that for any two networks $M_1, M_2$ having widths $w_1, w_2 > w$, $|\mathcal{R}(M_1, t) - \mathcal{R}(M_2, t)| \leq \epsilon$, where $\mathcal{R}(M, t)$ denotes the loss of network $M$ at time $t$.*

*Proof of Theorem 2.* The proof follows from the fact that the trajectory of the network approaches a limit as width tends to $\infty$, and thus, by definition of limits, there exists a width $w$, such that for any two networks with a width greater than $w$, their loss at time $t$ differs by at most $\epsilon$. $\qquad\square$

Note that the assumption of a fixed learning rate schedule with increasing width might seem strong, but recent works (Yang et al., 2022) have shown, that one of these initializations, termed as Maximal Update Parameterization ($\mu$P), exhibits hyperparameter transfer with width. This initialization scheme has also recently gained popularity because of this property and has been used by many open-source implementations (Dey et al., 2023a;b; Liu et al., 2023; Hu et al., 2024). Moreover, works (Yang et al., 2022; Vyas et al., 2023) have empirically demonstrated that with $\mu$P, networks start exhibiting consistent loss curves at practical widths.

Moreover, as the above theorem holds for a fixed batch size $B$ as well as $B \to \infty$, we expect that there exists a finite width $w$ such that the above theorem holds for all batch sizes $B$. **Thus, for fixed training tokens, we would expect that the critical batch size won't scale with model width beyond a point.** Although we have mostly talked about scaling model width, note that some recent results have also established such limits for infinite depth ResNets and transformers (Yang et al., 2024; Bordelon et al., 2024b;c), and thus the arguments above also hold for these networks.

## 4.2 FIXED MODEL SIZE AND SCALING UP DATA SIZE

We now turn to studying the impact of data size in mini-batch SGD for a well-specified Gaussian linear regression problem. Let $(\mathbf{x}, y)$ be a pair of covariates and responses from a population distribution. Let the population risk and the population distribution be

$$\mathcal{R}(\mathbf{w}) := \mathbb{E}(\mathbf{x}^\top \mathbf{w} - y)^2, \quad \mathbf{x} \sim \mathcal{N}(0, \mathbf{H}), \quad y|\mathbf{x} \sim \mathcal{N}(\mathbf{x}^\top \mathbf{w}^*, \sigma^2),$$

where $\mathbf{w}$ is the trainable parameter, the expectation is over the population distribution, and $(\mathbf{H}, \mathbf{w}^*, \sigma^2)$ specify the population distribution. Given $D$ independent samples $(\mathbf{x}_i, y_i)_{i=1}^D$ from the population distribution, we consider an estimate given by mini-batch SGD,

$$\mathbf{w}_0 = 0, \quad \mathbf{w}_{t+1} = \mathbf{w}_t - \gamma \frac{1}{B} \sum_{j=tB}^{(t+1)B-1} (\mathbf{x}_j^\top \mathbf{w}_t - y_j)\mathbf{x}_j, \quad t = 0, \ldots, n-1,$$

where $\gamma > 0$ is a constant learning rate, $B$ is the batch size, $n := D/B$ is the number of steps, $\mathbf{w}_0 = 0$ is the initialization (without loss of generality), and the output is the average of the iterates, $\bar{\mathbf{w}} := \frac{1}{n} \sum_{t=0}^{n-1} \mathbf{w}_t$. Then the following theorem provides a tight bound on the excess risk achieved by the average of the mini-batch SGD iterates.

We write $f(D) \lesssim g(D)$ if there is a positive constant $c$ such that $f(D) \leq cg(D)$ for every $D \geq 1$. We write $f(D) \asymp g(D)$ if $f(D) \lesssim g(D) \lesssim f(D)$. The proofs are all deferred to Appendix G.

**Theorem 3.** *Let $(\lambda_i)_{i>0}$ be the eigenvalues of $\mathbf{H}$ in nonincreasing order. Assume that $\|\mathbf{w}_0 - \mathbf{w}^*\|_{\mathbf{H}}^2 \lesssim \sigma^2$. Then for every $\gamma \lesssim \min\{B/\text{tr}(\mathbf{H}), 1/\|\mathbf{H}\|_2\}$, we have*

$$\mathbb{E}\mathcal{R}(\bar{\mathbf{w}}) - \sigma^2 \asymp \left(\frac{B}{D\gamma}\right)^2 \|\mathbf{w}_0 - \mathbf{w}^*\|_{\mathbf{H}_{0:k^*}^{-1}}^2 + \|\mathbf{w}_0 - \mathbf{w}^*\|_{\mathbf{H}_{k^*:\infty}}^2 + \sigma^2 \frac{k^* + (D\gamma/B)^2 \sum_{i>k^*} \lambda_i^2}{D},$$

*where $k^* := \max\{k : \lambda_k \geq B/(D\gamma)\}$ and the expectation is over the randomness of $\bar{\mathbf{w}}$.*

The proof of Theorem 3 is motivated by Zou et al. (2023). We focus on well-specified Gaussian data distribution for simplicity, but this can be relaxed to misspecified cases under fourth-moment conditions following the results in Zou et al. (2023). Theorem 3 suggests that for a fixed data size $D$, the excess risk depends on the batch size $B$ and the learning rate $\gamma$ only through their ratio $\gamma/B$. Moreover, a large $\gamma/B$ tends to decrease the bias error (the terms depending on $\mathbf{w}^*$) but increase the variance error (the terms depending on $\sigma^2$), and vice versa. This observation is exploited in the following corollary, where we compute the CBS that minimizes the sequential running time without sacrificing the rate of the attained excess risk.

**Corollary 2.** *Under the settings of [Theorem 3](#), additionally assume $\sigma^2 \asymp 1$ and the following capacity and source conditions:*

$$for\ a, b > 1: \quad \lambda_i \asymp i^{-a}, \quad \mathbb{E}\lambda_i \langle \mathbf{v}_i, \mathbf{w}_i^* \rangle^2 \asymp i^{-b}, \quad \mathbb{E}\lambda_i \langle \mathbf{v}_i, \mathbf{w}_i^* \rangle \langle \mathbf{v}_j, \mathbf{w}_j^* \rangle = 0\ for\ i \neq j,$$

*where $(\lambda_i, \mathbf{v}_i)_{i>0}$ are the eigenvalues and the corresponding eigenvectors of $\mathbf{H}$, and the expectation is over a prior of $\mathbf{w}^*$. Then we have*

1. *When $b \leq a$, the optimal hyper-parameters (that minimize the expected excess risk up to constant factors) are $\gamma^* \asymp 1$ and $B^* = 1$.*

2. *When $b > a$, the optimal hyper-parameters are $\gamma^*$ and $B^*$ such that*

$$0 < \gamma^* \lesssim 1, \quad 1 \leq B^* \leq D, \quad \gamma^*/B^* \asymp D^{\frac{a}{\min\{b,2a+1\}}-1}.$$

*Therefore, the CBS is $B^* \asymp D^{1-a/\min\{b,2a+1\}}$, which (along with $\gamma^* \asymp 1$) allows mini-batch SGD output $\bar{\mathbf{w}}$ to attain the optimal rate of the expected excess risk (as data size $D$ grows) with the smallest number of steps $n$.*

The capacity and source conditions are from the nonparametric linear regression literature (Caponnetto & De Vito, 2007) and are recently used to study scaling laws theory (Bordelon et al., 2024a; Lin et al., 2024; Paquette et al., 2024). According to [Corollary 2](#), when $b \leq a$, the bias error tends to dominate the variance error, in this case, the CBS is $B^* = 1$ to allow a maximum number of optimization steps. When $b > a$, the variance error tends to dominate the bias error, and the optimal choices of batch size and learning rate balance these two errors. While one can use $B^* = 1$ and a small $\gamma^*$ to achieve the best excess risk rate, this leads to a suboptimal sequential runtime ($n = D/B$). In this case, the CBS is $B^* \asymp D^{1-a/\min\{b,2a+1\}}$, which achieves the optimal excess risk rate while minimizing the sequential runtime.

---

**Takeaway on the theory of batch size scaling when scaling up data and model size:**

- As we scale up model size while keeping the data size fixed, $\mu P$ suggests that critical batch size does not scale with model width beyond a point.

- Fixing the model size, the critical batch size increases with the training duration. In the context of high-dimensional linear regression, where the variance error dominates the bias error, it is possible to choose a large batch size (as a function of the data size) for mini-batch SGD, allowing for reduced sequential runtime without compromising the rate at which excess risk is minimized.

---

## 5 RELATED WORK

**Scaling laws.** Scaling laws describe the parametric relationships among key factors involved in training neural networks: model size $N$, dataset size $D$, training cost $C$, and final training loss $\mathcal{R}$. These laws enable the prediction of training loss $\mathcal{R}$ based on available resources, making it possible to optimize resource allocation for efficient model training. For example, Hestness et al. (2017) found $\mathcal{R} \propto D^{-\alpha}$, with $\alpha \in [0.07, 0.35]$. Of the factors they varied, only tasks can change the exponent $\alpha$. Changing the architecture optimizers, regularizers, and loss functions, would only change the proportionality factor, not the exponent; Henighan et al. (2020) studied statistical relations between $N, D, C, \mathcal{R}$, over a wide range of values and found similar scaling laws, over the range of $N \in [10^3, 10^9], C \in [10^{12}, 10^{21}]$, and over multiple modalities (text, video, image, text to image, etc.). (Kaplan et al., 2020) states that $N$ should be scaled faster than $D$. However, Chinchilla scaling (Hoffmann et al., 2022a) found that models are under-trained, and then suggests that when given an increased budget (in FLOPs), to achieve compute-optimal, model size $N$ and data size $D$ should scale in approximately equal proportions. Recent efforts (Pearce & Song, 2024; Besiroglu et al., 2024; Porian et al., 2024) have been made in reproducing the scaling laws from (Hoffmann et al., 2022a) and the (Kaplan et al., 2020). Different from our focus on measuring the efficiency notion of CBS, most of them focus on deriving optimal hyper-parameters (Bi et al., 2024; Porian et al., 2024) including learning rate and batch size from small-scale training given a fixed compute budget FLOPs $\approx 6ND$ without decoupling the effects of model size and data size.

**Optimization and critical batch size.** Previous studies have shown that increasing batch sizes can be offset by a proportional adjustment to the learning rate in small-scale regimes (McCandlish et al., 2018; Zhang et al., 2019; Kaplan et al., 2020; Li et al., 2021). McCandlish et al. (2018) introduce the gradient noise scale, a measure that captures the variation in gradients across different training examples, which helps predict the critical batch size (CBS). Their findings also suggest that small-batch training is more compute-efficient, while large-batch training requires fewer optimizer steps. Momentum-based methods extend scaling to larger batch sizes but converge to the performance of standard SGD at smaller batch sizes (Shallue et al., 2019). Additionally, Zhang et al. (2019) analyze the impact of curvature on CBS using a noisy quadratic model, demonstrating that preconditioning techniques can increase the CBS. Golmant et al. (2018) show that the size of the dataset plays a smaller role in determining training efficiency compared to factors like model architecture and data complexity. In contrast, Hilton et al. (2022) examine how performance can be maintained at smaller batch sizes. Meanwhile, Smith et al. (2017); Smith & Le (2017) empirically investigated how the optimal learning rate changes based on momentum and training set size. Theoretical work has further sought to characterize CBS by analyzing SGD behavior in least-squares linear regression, especially in over-parameterized settings (Jain et al., 2018; Ma et al., 2018). Filatov et al. (2024) concurrently find that optimal batch size and CBS scale with data size. However, they do not explore how CBS scales with model size for models beyond 354M parameters, nor do they provide theoretical justifications or address the challenge of selecting optimal runs across a broad range of hyperparameters. Our work advances the optimization literature by formalizing CBS and quantifying its growth *w.r.t.* data size and emphasizing the importance of common hyper-parameter choices. It also provides strategies for studying large-scale pre-training beyond fixed training durations.

## 6 CONCLUDING REMARKS

In conclusion, this study provides an extensive examination of the scaling laws for critical batch size in large-scale autoregressive language model pre-training. By systematically analyzing the relationship between model size, data size, and CBS, we found that while CBS increases with data size, it remains relatively invariant to model size. This finding suggests training on more data may enable greater data parallelism in pre-training. We further emphasize the role of key hyperparameters and exponential weight averaging, which can match the performance of cosine scheduling without requiring fixed training durations. These insights offer practical strategies for scaling models while maintaining efficiency, which is critical in resource-constrained scenarios.

## ACKNOWLEDGEMENTS

We thank Dhruv Madeka. We thank Qirong Ho for discussions on distributed training; Jeremy Berstein for discussions on optimization; Dirk Groeneveld and Luca Soldaini for discussions on OLMo implementations. HZ is supported by an Eric and Susan Dunn Graduate Fellowship. SK and DM acknowledge the Chan Zuckerberg Initiative Foundation to establish the Kempner Institute for the Study of Natural and Artificial Intelligence; SK acknowledges the support from the Office of Naval Research under award N00014-22-1-2377, and the National Science Foundation Grant under award #IIS 2229881. NV and DM are supported by a Simons Investigator Fellowship, NSF grant DMS-2134157, DARPA grant W911NF2010021, and DOE grant DE-SC0022199.

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

# Appendix

## Table of Contents

# A  TRAINING DYNAMICS

**A simple strategy for setting warmup steps.** To further prove that the critical batch size actually exists and the saturation of large batch sizes is not an artifact of not training well with proper hyper-paragrams, we take into account the warmup fraction in training as well:

We sweep over warmup step ratios (how many fraction of training steps do we need to linearly scale the learning rate from zero) over 0.25 and 0.1 and find that 0.25 works best for 85M models. Therefore, we fix this number of warmup steps to be 0.25 of the $t_{\text{Chin}}$ for future experiments. For 151M models, we sweep over the fraction of warmup steps in $\{0.15, 0.25, 0.35\}$. We show in Figure 7 that using a warmup ratio of **0.25** can be a reasonable design choice as it enjoys consistently better performance than 0.15 yet only slightly underperforms 0.35. After we find that setting the warmup steps according to this heuristic, we use the ratio proportionally for all the other model sizes. This strategy has also been shown to be effective in (Porian et al., 2024).

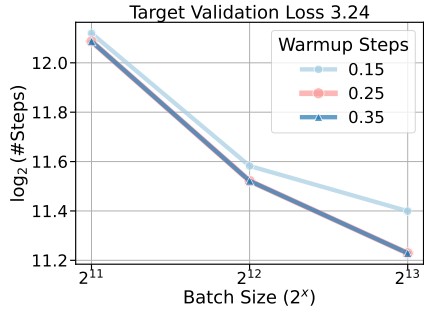

Figure 7: Ablation of warmup steps used in the linear LR warmup stage for large batch sizes.

**Examining the last part of training.** By closely examining the final stages of the training process (Figure 8), it becomes apparent that applying Exponentially Weighted Averages (EWA) can help smooth out noise, allowing the optimization to converge to the target loss more efficiently. For example, a very high EWA decay rate would be needed even for a 1.2B model with a moderate batch size of 1024. Moreover, we observe that the optimization process is notably influenced by the final phase of training. For instance, by step 10,000, most runs achieve a validation loss below 3.2 (Figure 8a), and similarly, a loss below 2.8 is reached by step 30,000 (Figure 8b). However, to reach the target loss of 2.736, the difference between the best and second-best runs grows substantially, with the best run requiring over 5,000 fewer steps.

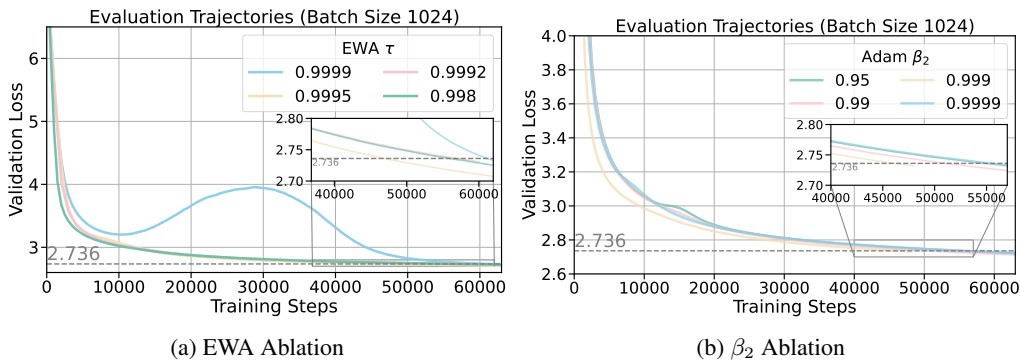

(a) EWA Ablation                    (b) $\beta_2$ Ablation

Figure 8: **A large enough EWA decay rate $\tau$ and Adam $\beta_2$ is important for long-duration training.** We plot the evaluation curves of 1.2B models, as in Chinchilla settings, we scale up data size proportionally to model size. When increasing the number of training tokens, it is crucial to carefully set appropriate values for both $\beta_2$ and $\tau$ to effectively account for efficiency.

**Scheduler comparison for other batch sizes**. In Figure 9, we include more comparisons on different schedulers that are reported in the main text (Figure 1). Overall, our Constant+EWA performs competitively with cosine scheduling and outperforms WSD scheduling, especially for large batch size regimes. Note that we sweep over the decay steps as $0.1, 0.2, 0.3\times$ total training steps for WSD scheduling. We tune cosine scheduling by conducting sweeps over various maximum optimization steps to identify the optimal value, and then rerun the training using this step count. This approach ensures that the model reaches the target loss near the end of training, optimizing the performance of learning rate decay. For schedule-free optimizers, we tune the $\beta_1$ 0.9, 0.95, 0.98. Under small

batch sizes, the schedule-free optimizer (Defazio et al., 2024) is a competitive baseline but it is significantly worse for batch sizes larger than 1024.

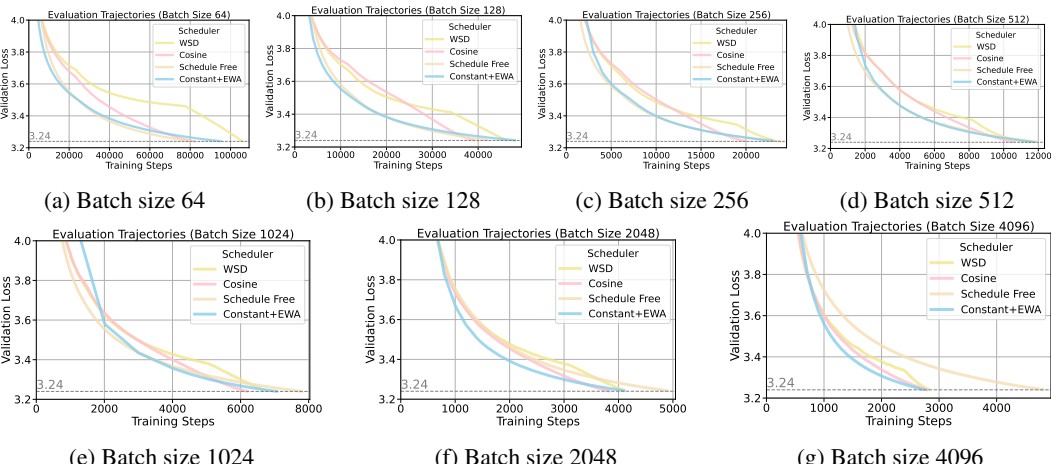

(a) Batch size 64  (b) Batch size 128  (c) Batch size 256  (d) Batch size 512

(e) Batch size 1024  (f) Batch size 2048  (g) Batch size 4096

Figure 9: Scheduler comparison for different batch sizes. All are with model size 151M.

**Longer training requires higher EWA decay rate $\tau$.** Throughout the paper, we adopt a learning rate of $0.00316$ across most experiments, but it is unclear whether that would be sub-optimal, especially since training with longer duration may require a lower learning rate as suggested in (DeepSeek-AI et al., 2024). Therefore, we justify our design decision on tuning EWA decay rate for simulating learning rate decay on different training durations by conducting the following experiments on a series of 151M models with (a) batch size 256, $0.5\times$ Chinchilla tokens; (b) batch size 256, $20\times$ Chinchilla tokens; (c) batch size 2048, $20\times$ Chinchilla tokens, all with learning rate swept over $\{0.00316, 0.00158, 0.01264, 0.00632, 0.00075\}$ and EWA decay rate $\tau$ over $\{0.99, 0.9968, 0.999, 0.99968, 0.9999\}$. We set the number of warmup steps to be $0.25$ of the total steps for (a) and $0.05$ for (b) and (c). The results in Figure 10 denote the validation loss at the end of the training, which consistently show that within each group of experiments, a learning rate of $0.00316$ we use throughout the paper is consistently the best. Moreover, when enlarging the training data size from $0.5\times$ Chinchilla tokens to $20\times$, the optimal EWA decay rate value $\tau$ would increase as well. This is also justified in the results presented in Figure 8 and Table 4 which indicate that longer training durations may benefit from a higher EWA decay rate to improve optimization performance.

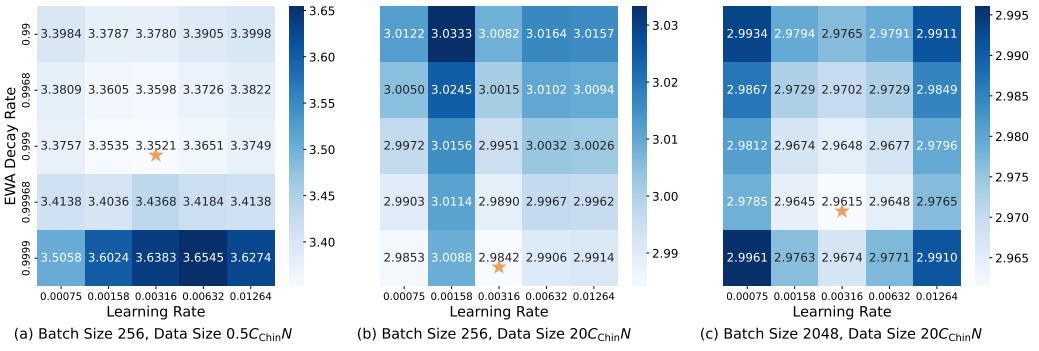

(a) Batch Size 256, Data Size $0.5C_{Chin}N$  (b) Batch Size 256, Data Size $20C_{Chin}N$  (c) Batch Size 2048, Data Size $20C_{Chin}N$

Figure 10: Impact of learning rate and EWA decay rate across various training durations. We report the validation loss at the end of training for each hyper-parameter combination. $N$ denotes the model size. The best loss is marked by a ★ symbol. Longer training durations, as seen in (b) and (c), necessitate a higher EWA decay rate for a given learning rate.

**Effects of weight decay.** Though weight decay (WD) does not provide generalization bene-fits for pre-training, previous works have shown that it might improve convergence in language model training (Hoffmann et al., 2022a; Kosson et al., 2023). We adopt the default decoupled weight decay (Loshchilov, 2017) implementation in PyTorch and sweep over weight decay rate $\{0.01, 0.0316, 0.1\}$ for LR 0.01, $\{0.0316, 0.1, 0.316\}$ for LR 0.00316. We show in Figure 11 that for constant learning rate with EWA, while weight decay offers a slight performance improvement, it has little effect on the critical batch size, which remains our primary focus. Consequently, we disable weight decay throughout the paper.

# B ADDITIONAL ABLATION STUDIES ON STUDYING ADAM OPTIMIZER

We employ Adam as the default optimizer for large-scale model training throughout the paper. In this section, we focus on two key hyper-parameters that sig-nificantly affect optimization efficiency and examine their im-pact in detail.

**The effect of momentum $\beta_1$ of Adam on CBS**. We sweep over several momentum $\beta_1$ values in Adam for all learning rates and batch sizes: $[0, 0.8, 0.9, \mathbf{0.95}, 0.975]$. Overall, Figure 12 shows that language model pre-training may need a large momentum value to be efficient and $\beta_1 = 0.95$ is slightly better ($<0.02$ gain on eval loss) than 0.9 for batch sizes. We observe that in small batch size regimes like $2^6$, the performance gap between opti-mizing with and without momentum $\beta_1$ is small while the gap increases as we double the batch size (Shallue et al., 2019). Moreover, we show that momentum 0.9 and 0.975 have simi-lar effects on the number of steps needed to reach a target val-idation loss and critical batch sizes. On the other hand, small momentum, especially no momentum, would hurt the optimiza-tion. This aligns well with the extensively studied acceleration of momentum in SGD with momentum (Goh, 2017).

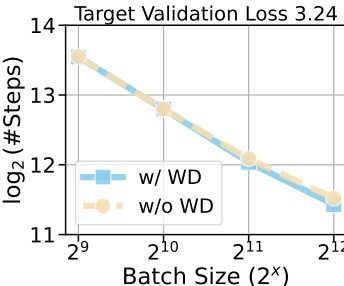

Figure 11: Comparison of effi-ciency with and without weight decay

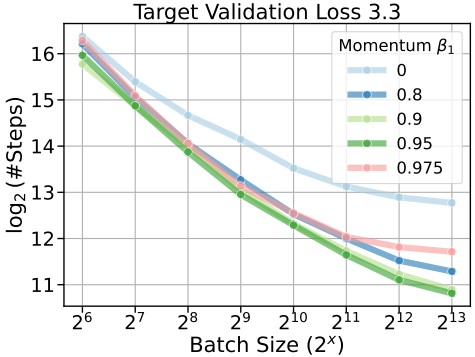

Figure 12: Ablation results on **momentum**. All data points are trained using 151M models and a total number of optimization steps to reach a fixed target loss is reported.

**The effect of the second moment decay rate $\beta_2$ in Adam**. As reported in Appendix Table 4, we found that $\beta_2$ in Adam, the exponential decay rate of the second momentum estimate of gra-dients to smooth out the model update, also has significant effects on training for small batch sizes. This might be because gradients in small-batch training are sparser. Specifically, we ab-late $\beta_2 \in [0.95, 0.99, 0.999]$ for all model sizes and batch sizes in $[64, 128, 256, 512]$. We find that the default value 0.95 in previous works that are set for millions of tokens batch size training might be sub-optimal (Smith et al., 2022; Wortsman et al., 2023; Groeneveld et al., 2024). For large batch sizes $[1024, 2048, 4096, 8192]$, we experiment with a small $\beta_2 = 0.9$ with the model size 151M, finding that it is worse than the default 0.95 we choose. When training a larger model with a longer duration (e.g. Chinchilla settings in Appendix Figure 8b), a high enough $\beta_2$ is necessary.

> **Takeaway on Adam optimizer:**
>
> - Momentum $\beta_1$ is important in improving training efficiency: a value of 0.95 consistently performs well across various model sizes and batch sizes. However, setting it too high (0.975) or too low (0.8) leads to sub-optimal results.
>
> - Smaller $\beta_2 = 0.95$ is helpful for large-batch training over short durations, while a large $\beta_2 = 0.99, 0.999$ or $0.9995$ is helpful for long-duration training and substantially improves small batch size training ($<$262k tokens).

## C    RESULTS INCLUDING SMALL BATCH SIZES

For completeness, we demonstrate linear scaling behavior in small-batch regimes across all model sizes (Figure 13). This shows that all models exhibit linear scaling (with reasonable deviations) with a batch size ranging from $2^6$ to $2^{10}$, where doubling the batch size roughly halves the number of steps needed to reach a target validation loss, as determined by the optimal run with a batch size of 256 at the Chinchilla step.

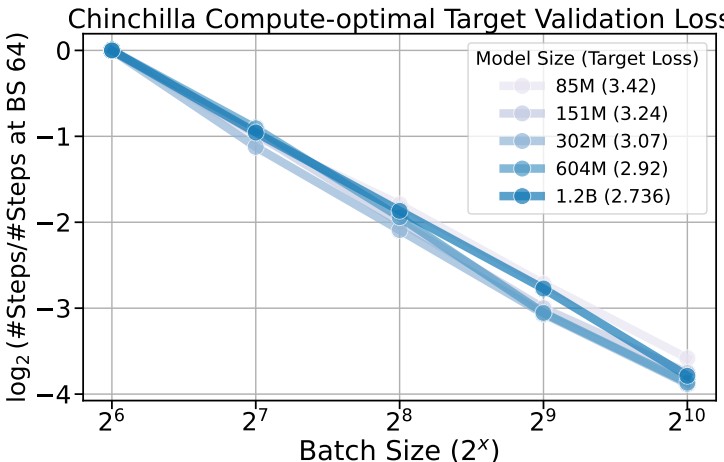

Figure 13: **Linear scaling regimes**: doubling the batch size can halve the optimization steps to reach the target loss.

Moreover, we include all the results that contain the smallest several batch sizes (Figure 14). Note that the denominator is the number of steps to reach target loss at batch size **64** instead of 256 now. Now we can observe clear linear scaling of all the model sizes till around $2^{10}$ for model sizes, while the largest three model sizes maintain linear scaling till almost $2^{11}$. There are minor differences with the main plot in Figure 1 because of the difficulty of optimizing with very small batch sizes like 64 but it does not affect the conclusions and takeaways we would like to convey. Since our focus is primarily on large batch sizes, we consistently use $2^9$ as the starting batch size throughout the main text. Furthermore, recall that we set $B_{\text{opt}} = 2^8$ when selecting the target loss. Figure 13 and Figure 14 confirm that $2^8$ falls within the linear scaling regime, which justifies our design choice.

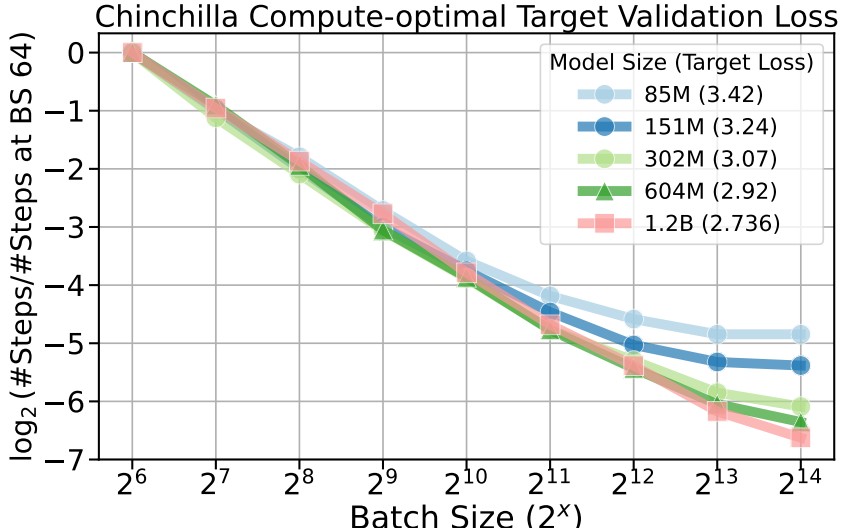

Figure 14: **Full results for different models in Chinchilla settings**. We include both a largest batch size $2^{14}$ start the plot from several small batch sizes $2^6, 2^7, 2^8$. Relative number of steps *w.r.t.* batch size $\mathbf{2^6}$ is reported.

# D    ADDITIONAL DETAILS ON EXPERIMENTS

**Optimizer setup**. For optimizers, we try both SGD (Robbins & Monro, 1951) and Adam (Kingma, 2014) and find that SGD without momentum is significantly worse so we use Adam only for all the experiments. We disable weight decay in Adam as we observe that it does not significantly affect critical batch size (Figure 11). For generality, though the training set C4 might contain low-quality or duplicated documents that can potentially lead to training instability (Muennighoff et al., 2023; Wortsman et al., 2023), we observed that these issues did not affect our primary target of interest—namely, the final optimization efficiency. As a result, we didn't explicitly adopt additional normalization like QK normalization (Dehghani et al., 2023; Zhai et al., 2023) or a z-loss (Chowdhery et al., 2022) to mitigate loss spikes.[2] We set $\epsilon$ to be 1e-8 by default, and refer to momentum as $\beta_1$ in Adam by default throughout the paper.

**Chinchilla steps for batch size 256 that determine the target losses.** For each model size, we aim to establish a target validation loss by training with a global batch size of 256 on a Chinchilla-optimal amount of tokens. Given the context length of 512 used throughout, we can determine the number of training steps required based on the following Table 1. We use a token-to-model size ratio $C_{\text{Chin}}$ of approximately 20.34 to study the halving effects of doubling the batch size and to observe its impact on the critical batch size.

Table 1: Chinchilla steps for determining the target loss for each model size.

| Model Size | 85M | 151M | 302M | 604M | 1.2B |
|---|---|---|---|---|---|
| Chinchilla Step | 13193 | 23438 | 46875 | 93750 | 187500 |

**Evaluation data size and frequency**.    To ensure frequent model evaluation on a holdout C4 validation set, it is important to maintain a balance between reliability and efficiency. A larger evaluation set size can provide more stable and reliable performance metrics, but it must also be efficient to maintain practicality in each run. Using 151M models, we evaluated the variance across different token counts: 2.17e-4 for 327,680 tokens, 4.53e-5 for 1,638,400 tokens, and 7.65e-6 for 3,276,800 tokens. Based on these results, we have set the default number of evaluation batches to

---

[2]We observe irregular loss spikes despite adopting gradient clipping, but most runs can still be optimized well in the end.

100. It is important to note that the total number of training steps varies across different batch sizes. To address this, we implement a hybrid evaluation protocol: the model is evaluated at intervals of $2^i$ (where $i \in \mathbb{Z}$), every 1,000 steps, and at $0.7n, 0.75n, 0.8n, \ldots$, up to $n$ during the last 30% of the total steps $n$. This approach ensures more frequent evaluations toward the end of training, allowing for a more accurate assessment of the total training steps needed to achieve a target evaluation loss.

**Hyper-parameter search details**. Due to compute constraints, we cannot perform an exhaustive search over all hyper-parameter configurations. Instead, as suggested by the ablation studies in the main text, we gain insights into hyper-parameters by training smaller proxy models (151M parameters). We optimize the following hyper-parameters in sequence: learning rate, momentum ($\beta_1$), warmup steps, scheduler, and context length. Additionally, we tune $\beta_2$ and $\tau$ for each model size and batch size. Specifically, for large batch sizes ($>1024$), a smaller $\beta_2$ and larger $\tau$ tend to be more effective, while the opposite holds true for smaller batch sizes, aligning with findings in (Porian et al., 2024; Zhang et al., 2022).

Below we show the hyper-parameter choices (Table 3) and optimal ones (Table 4) we report in our main plot for studying CBS with respect to model sizes. Additionally, Table 5 presents various model size configurations and scaling methods, with models in **bold** indicating those used in our controlled experiments.

Table 2: Model architecture details.

| Model Size | $n_{\text{heads}}$ | $n_{\text{layers}}$ | $d_{\text{model}}$ | Hidden size of MLPs |
|---|---|---|---|---|
| 85M | 12 | 12 | 768 | 3072 |
| 151M | 16 | 12 | 1024 | 4096 |
| 302M | 16 | 24 | 1024 | 4096 |
| 604M | 16 | 12 | 2048 | 8192 |
| 1.2B | 32 | 24 | 2048 | 8192 |

Table 3: Sweeping experiments settings. Default values after the hyper-parameter search are in **Bold** font. Bold font means the default hyper-parameters that can closely reproduce our results without extensive tuning. Not bolding implies a full sweep for each model scale. The values in parentheses were not used for every sweep: for the 151M models, we tested learning rates of 3.16e-4 and 1e-2, but found that with EWA, these performed worse than 3.16e-3. The EWA decay rate of 0.99995 is only used for long 1.2B runs.

| Hyper-parameter | Values |
|---|---|
| Model Size | 85M, 151M, 302M, 604M, 1.2B |
| Batch size | $2^6 \sim 2^{14}$ |
| Learning rate | (3.16e-4), 1e-3, **3.16e-3**, (1e-2) |
| Learning rate scheduler | **constant+EWA**, cosine, WSD, schedule free |
| Warmup fraction | 0.15, **0.25**, 0.35 |
| Momentum $\beta_1$ | 0, 0.8, 0.9, **0.95**, 0.975 |
| Adam $\beta_2$ | 0.95, 0.99, 0.995, 0.999, 0.9995 |
| EWA decay rate $\tau$ | 0.95, 0.98, 0.99, 0.995, 0.998, 0.999, 0.9995, (0.99995) |
| Context Length | **512**, 1024, 2048, 4096 |
| Grad clipping norm | 1.0 |

Table 4: **Optimal** hyper-parameters for different model sizes. The optimal means the number of steps to reach a target validation loss. We refer $\beta_2$ as the exponential decay rate for the second-moment estimates in Adam, $\tau$ as the interpolation parameters in EWA ($\xi_{t+1} = \tau \cdot \xi_t + (1 - \tau) \cdot \theta_t$). All the optimal runs are trained with momentum $\beta_1 = 0.95$ and learning rate 3.16e-3.

| Batch Size | $\beta_2$ | $\tau$ | Batch Size | $\beta_2$ | $\tau$ |
|---|---|---|---|---|---|
| | 85M | | | 151M | |
| 64 | 0.999 | 0.9995 | 64 | 0.99 | 0.998 |
| 128 | 0.999 | 0.9995 | 128 | 0.99 | 0.998 |
| 256 | 0.999 | 0.9995 | 256 | 0.99 | 0.998 |
| 512 | 0.999 | 0.998 | 512 | 0.99 | 0.998 |
| 1024 | 0.95 | 0.99 | 1024 | 0.95 | 0.95 |
| 2048 | 0.95 | 0.99 | 2048 | 0.99 | 0.99 |
| 4096 | 0.95 | 0.99 | 4096 | 0.99 | 0.99 |
| 8192 | 0.95 | 0.98 | 8192 | 0.95 | 0.95 |
| 16384 | 0.95 | 0.98 | 16384 | 0.99 | 0.99 |

| Batch Size | $\beta_2$ | $\tau$ | Batch Size | $\beta_2$ | $\tau$ | Batch Size | $\beta_2$ | $\tau$ |
|---|---|---|---|---|---|---|---|---|
| | 302M | | | 604M | | | 1.2B | |
| 64 | 0.999 | 0.9995 | 64 | 0.9995 | 0.9995 | 64 | 0.999 | 0.9995 |
| 128 | 0.999 | 0.9995 | 128 | 0.9995 | 0.9995 | 128 | 0.999 | 0.9995 |
| 256 | 0.995 | 0.9995 | 256 | 0.9995 | 0.9995 | 256 | 0.995 | 0.9995 |
| 512 | 0.99 | 0.9995 | 512 | 0.9995 | 0.9995 | 512 | 0.99 | 0.9995 |
| 1024 | 0.99 | 0.999 | 1024 | 0.999 | 0.999 | 1024 | 0.99 | 0.999 |
| 2048 | 0.95 | 0.998 | 2048 | 0.998 | 0.998 | 2048 | 0.95 | 0.998 |
| 4096 | 0.95 | 0.995 | 4096 | 0.995 | 0.995 | 4096 | 0.95 | 0.995 |
| 8192 | 0.95 | 0.99 | 8192 | 0.99 | 0.99 | 8192 | 0.95 | 0.99 |
| 16384 | 0.99 | 0.99 | 16384 | 0.99 | 0.995 | 16384 | 0.95 | 0.99 |

Table 5: Model architectures of the ablation study on the scaling of Depth and Width. Only models highlighted in **bold** are used, as they are more comparable in terms of model size.

| Model Size | $n_{\text{heads}}$ | $\mathbf{n_{layers}}$ | $d_{\text{model}}$ | $\text{MLP}_{\text{hidden}}$ |
|---|---|---|---|---|
| **151M** | 16 | 12 | 1024 | 4096 |
| 302.09M | 16 | 24 | 1024 | 4096 |
| **604.18M** | 16 | 48 | 1024 | 4096 |
| 1.208B | 16 | 96 | 1024 | 4096 |

| Model Size | $n_{\text{heads}}$ | $n_{\text{layers}}$ | $\mathbf{d_{model}}$ | $\mathbf{MLP_{hidden}}$ |
|---|---|---|---|---|
| **151M** | 16 | 12 | 1024 | 4096 |
| 339.81M | 24 | 12 | 1536 | 6144 |
| **604.08M** | 32 | 12 | 2048 | 8192 |
| 943.84M | 40 | 12 | 2560 | 10240 |

## E  ADDITIONAL DETAILS ON SCALING LAWS

We first present the fitted power law relationship between the number of optimization steps required to reach the target loss and the batch size (Table 6). All the results are obtained by solving the equation in Section 3.2 via `scipy.optimize.fsolve` using default hyper-parameters.

We report forecasting results for various model and token sizes, extending beyond the plots presented in the main text (Table 7). For each row, increasing either model size or token size shows that the forecasting results remain comparable.

Note that our definition of CBS and its scaling law have a similar interpretation with the one in (McCandlish et al., 2018; Kaplan et al., 2020) as $\frac{E_{\min}}{S_{\min}}$, $S_{\min}$ denotes the minimum possible number

Table 6: Fitted scaling law parameters for Chinchilla settings when fixing $\alpha = 1$: $\log(Y) = \log(a + \frac{b}{B^\alpha})$, where $Y$ is the number of steps to reach Chinchilla target loss, $B$ denotes the batch size, and the critical batch size is solved as $B^* = (\frac{b+5a \times 1.2 \times B_{\text{opt}}}{5a})^{\frac{1}{\alpha}}$, $B_{\text{opt}} = 256$.

(a) Fixed $\alpha = 1$ (default)

| Model Size | $a$ | $b$ | $\alpha$ | $\log_2(B^*)$ |
|---|---|---|---|---|
| 85M | 1293.83 | 2834258.08 | 1 | 9.54 |
| 151M | 1752.42 | 5677478.78 | 1 | 9.90 |
| 302M | 2095.35 | 11383269.89 | 1 | 10.44 |
| 604M | 2459.93 | 19449688.59 | 1 | 10.88 |
| 1.2B | 3897.31 | 43381130.22 | 1 | 11.31 |

(b) Fitted $\alpha$

| Model Size | $a$ | $b$ | $\alpha$ | $\log_2(B^*)$ |
|---|---|---|---|---|
| 85M | 1348.31 | 3386537.23 | 1.03 | 9.34 |
| 151M | 1943.53 | 8259867.95 | 1.07 | 9.51 |
| 302M | 2281.48 | 13977184.09 | 1.04 | 10.20 |
| 604M | 2733.81 | 23738850.26 | 1.04 | 10.62 |
| 1.2B | 3388.53 | 36748556.71 | 0.97 | 11.62 |

Table 7: Additional forecasted CBS results for larger scale. Recall that we fit $B^* = 93.20 \times N^{0.47}$, $B^* = 22.91 \times D^{0.47}$ where model size $N$ is in millions and data size $D$ is in billions.

| Model Size | Forecasted CBS | $\log_2(B^*)$ | Token Size | Forecasted CBS | $\log_2(B^*)$ |
|---|---|---|---|---|---|
| 1.5B | 2862.17 | 11.48 | 30B | 2833.31 | 11.47 |
| 2B | 3274.93 | 11.68 | 40B | 3240.99 | 11.66 |
| 2.5B | 3635.65 | 11.83 | 50B | 3597.20 | 11.81 |
| 3B | 3959.69 | 11.95 | 60B | 3917.12 | 11.94 |
| 3.5B | 4256.09 | 12.06 | 70B | 4209.70 | 12.04 |
| 4B | 4530.72 | 12.15 | 80B | 4480.76 | 12.13 |
| 4.5B | 4787.63 | 12.23 | 90B | 4734.29 | 12.21 |
| 5B | 5029.77 | 12.30 | 100B | 4973.22 | 12.28 |
| 5.5B | 5259.34 | 12.36 | 110B | 5199.73 | 12.34 |
| 6B | 5478.06 | 12.42 | 120B | 5415.52 | 12.40 |

of steps taken to reach target loss and $E_{\min}$ is the minimum possible number of training examples processed to reach target loss. In particular, recall that critical batch size can be analytically derived as $B^* = \frac{b}{5a} + 1.2B_{\text{opt}}$. This relationship reflects the point where batch size scaling incurs a 20% overhead when the batch size is doubled while (McCandlish et al., 2018). Here, the parameter $b$ plays a role analogous to $E_{\min}$, while $a$ corresponds to $S_{\min}$, depending on the specific overhead chosen to characterize the diminishing returns from increasing the batch size. We also note that the diminishing return overhead can vary, leading to the following observations: $10\% : B^* = 20.67 \times D^{0.48}, 20\% : B^* = 22.91 \times D^{0.47}, 50\% : B^* = 30.50 \times D^{0.44}$.

## F    REPRODUCIBILITY

In our training environment, we verify that, across multiple model sizes (2.4M, 9.4M, 19M, 42M, 85M, 151M, 302M), we can (approximately) reproduce the final evaluation loss of Figure 1 in (Wortsman et al., 2023). We use nodes equipped with 8 A100 GPUs, each with 80GiB of memory, for model training. We built our training framework using the Olmo training suite (Groeneveld et al., 2024).

## G    COMPLETE PROOFS IN SECTION 4.2

*Proof of Theorem 3.* The work by Zou et al. (2023) studied SGD with batch size 1 for linear regression and established matching (up to a constant factor) upper and lower bounds on the excess risk. Our theorem generalizes theirs by further considering the effect of batch size. Our analysis uses their intermediate results through appropriate reductions. We first define a set of operations on PSD matrices as follows:

$$\mathcal{I} = \mathbf{I} \otimes \mathbf{I}, \quad \mathcal{M}^B = \mathbb{E}\left[\left(\frac{1}{B}\sum_{i\in\mathcal{I}}\mathbf{x}_i\mathbf{x}_i^\top\right) \otimes \left(\frac{1}{B}\sum_{i\in\mathcal{I}}\mathbf{x}_i\mathbf{x}_i^\top\right)\right], \quad \widetilde{\mathcal{M}} = \mathbf{H} \otimes \mathbf{H},$$

$$\mathcal{T}^B = \mathbf{H} \otimes \mathbf{I} + \mathbf{I} \otimes \mathbf{H} - \gamma\mathcal{M}^B, \quad \widetilde{\mathcal{T}} = \mathbf{H} \otimes \mathbf{I} + \mathbf{I} \otimes \mathbf{H} - \gamma\mathbf{H} \otimes \mathbf{H},$$

where $\mathcal{I}$ is an index set of $B$ independent data. Note that

$$\left(\mathcal{M}^{\mathrm{B}} - \widetilde{\mathcal{M}}\right) \circ \mathbf{A} = \mathrm{Cov}\left(\frac{1}{B} \sum_{i \in \mathcal{I}} \mathbf{x}_i \mathbf{x}_i^\top \mathbf{A}^{1/2}\right) = \frac{1}{B} \mathrm{Cov}(\mathbf{x}\mathbf{x}^\top \mathbf{A}^{1/2}).$$

For Gaussian data $\mathbf{x} \in \mathcal{N}(0, \mathbf{H})$, we have

$$\mathrm{Cov}(\mathbf{x}\mathbf{x}^\top \mathbf{A}^{1/2}) = \mathbb{E}_{\mathbf{x} \in \mathcal{N}(0,\mathbf{H})}\left[\mathbf{x}\mathbf{x}^\top \mathbf{A}\mathbf{x}\mathbf{x}^\top\right] - \mathbf{H}\mathbf{A}\mathbf{H} = 2\mathrm{tr}(\mathbf{H}\mathbf{A})\mathbf{H}.$$

Together, we obtain

$$\left(\mathcal{M}^{\mathrm{B}} - \widetilde{\mathcal{M}}\right) \circ \mathbf{A} = \frac{2}{B} \mathrm{tr}(\mathbf{H}\mathbf{A})\mathbf{H}.$$

Now we compute the error propagation along the SGD steps. Let $\boldsymbol{\eta}_t = \mathbf{w}_t - \mathbf{w}^*$ be the error vector. For convenience, let $\mathbf{G}_t = \frac{1}{B} \sum_{i \in \mathcal{I}_t} \mathbf{x}_i \mathbf{x}_i^\top$ be the empirical covariance of an independent batch. Then we can define the bias and variance iterates as

$$\boldsymbol{\eta}_t^{\mathrm{bias}} = \left(\mathbf{I} - \gamma \mathbf{G}_t\right)\boldsymbol{\eta}_{t-1}^{\mathrm{bias}}, \quad t = 1, \ldots, n-1, \quad \boldsymbol{\eta}_0^{\mathrm{bias}} = \mathbf{w}_0 - \mathbf{w}^*,$$

and

$$\boldsymbol{\eta}_t^{\mathrm{variance}} = \left(\mathbf{I} - \gamma \mathbf{G}_t\right)\boldsymbol{\eta}_{t-1}^{\mathrm{variance}} + \gamma \cdot \frac{1}{B} \sum_{i \in \mathcal{I}_t} \xi_i \mathbf{x}_i, \quad t = 1, \ldots, n-1, \quad \boldsymbol{\eta}_0^{\mathrm{variance}} = \mathbf{0},$$

where $\xi_i = y_i - \mathbf{x}_i^\top \mathbf{w}^* \sim \mathcal{N}(0, \sigma^2)$. We then compute the covariance matrices of these two error iterates

$$\mathbf{B}_t^{\mathrm{B}} := \mathbb{E}[\boldsymbol{\eta}_t^{\mathrm{bias}} \otimes \boldsymbol{\eta}_t^{\mathrm{bias}}], \quad \mathbf{C}_t^{\mathrm{B}} := \mathbb{E}[\boldsymbol{\eta}_t^{\mathrm{variance}} \otimes \boldsymbol{\eta}_t^{\mathrm{variance}}].$$

Using the operators, these covariance matrices take the following iterative updates:

$$\mathbf{B}_0^{\mathrm{B}} = \boldsymbol{\eta}_0 \otimes \boldsymbol{\eta}_0, \quad \mathbf{B}_t^{\mathrm{B}} = \mathbb{E}_{\mathbf{G}_t}\left[(\mathbf{I} - \gamma \mathbf{G}_t)\mathbf{B}_{t-1}^{\mathrm{B}}(\mathbf{I} - \gamma \mathbf{G}_t)\right] = \left(\mathcal{I} - \gamma \mathcal{T}^{\mathrm{B}}\right) \circ \mathbf{B}_{t-1}^{\mathrm{B}},$$

$$\mathbf{C}_0^{\mathrm{B}} = \mathbf{0}, \quad \mathbf{C}_t^{\mathrm{B}} = \mathbb{E}_{\mathbf{G}_t}\left[(\mathbf{I} - \gamma \mathbf{G}_t)\mathbf{C}_{t-1}^{\mathrm{B}}(\mathbf{I} - \gamma \mathbf{G}_t)\right] + \frac{\gamma^2}{B^2}\mathbb{E}\left[\left(\sum_{i \in \mathcal{I}_t} \xi_i \mathbf{x}_i\right)\left(\sum_{i \in \mathcal{I}_t} \xi_i \mathbf{x}_i\right)^\top\right]$$

$$= \left(\mathcal{I} - \gamma \mathcal{T}^{\mathrm{B}}\right) \circ \mathbf{C}_{t-1}^{\mathrm{B}} + \frac{\gamma^2 \sigma^2}{B}\mathbf{H},$$

where the last equation is because

$$\mathbb{E}\left[\left(\sum_{i \in \mathcal{I}_t} \xi_i \mathbf{x}_i\right)\left(\sum_{i \in \mathcal{I}_t} \xi_i \mathbf{x}_i\right)^\top\right] = \mathbb{E}\left[\sum_{i \in \mathcal{I}_t} \xi_i^2 \mathbf{x}_i \mathbf{x}_i^\top\right] = \sigma^2 B \mathbf{H}.$$

Recall that $\bar{\mathbf{w}} = \frac{1}{n} \sum_{t=0}^{n-1} \mathbf{w}_t$. First, using Lemmas B.3 and C.1 in (Zou et al., 2023), we get the following bias-variance decomposition (note that our setting is well-specified):

$$\mathbb{E}[\mathcal{R}(\bar{\mathbf{w}})] - \min \mathcal{R}(\cdot) = \mathrm{bias} + \mathrm{variance},$$

where

$$\mathrm{bias} := \frac{1}{2}\langle \mathbf{H}, \mathbb{E}[\bar{\boldsymbol{\eta}}^{\mathrm{bias}} \otimes \bar{\boldsymbol{\eta}}^{\mathrm{bias}}]\rangle \begin{cases} \leq \dfrac{1}{n^2} \displaystyle\sum_{t=0}^{n-1}\sum_{k=t}^{n-1} \left\langle (\mathbf{I} - \gamma \mathbf{H})^{k-t}\mathbf{H}, \mathbf{B}_t^{\mathrm{B}}\right\rangle, \\[3mm] \geq \dfrac{1}{2n^2} \displaystyle\sum_{t=0}^{n-1}\sum_{k=t}^{n-1} \left\langle (\mathbf{I} - \gamma \mathbf{H})^{k-t}\mathbf{H}, \mathbf{B}_t^{\mathrm{B}}\right\rangle, \end{cases}$$

$$\mathrm{variance} := \frac{1}{2}\langle \mathbf{H}, \mathbb{E}[\bar{\boldsymbol{\eta}}^{\mathrm{variance}} \otimes \bar{\boldsymbol{\eta}}^{\mathrm{variance}}]\rangle \begin{cases} \leq \dfrac{1}{n^2} \displaystyle\sum_{t=0}^{n-1}\sum_{k=t}^{n-1} \left\langle (\mathbf{I} - \gamma \mathbf{H})^{k-t}\mathbf{H}, \mathbf{C}_t^{\mathrm{B}}\right\rangle, \\[3mm] \geq \dfrac{1}{2n^2} \displaystyle\sum_{t=0}^{n-1}\sum_{k=t}^{n-1} \left\langle (\mathbf{I} - \gamma \mathbf{H})^{k-t}\mathbf{H}, \mathbf{C}_t^{\mathrm{B}}\right\rangle, \end{cases}$$

where

$$\bar{\boldsymbol{\eta}}^{\text{bias}} := \frac{1}{n} \sum_{t=0}^{n-1} \bar{\boldsymbol{\eta}}_t^{\text{bias}}, \quad \bar{\boldsymbol{\eta}}^{\text{variance}} := \frac{1}{n} \sum_{t=0}^{n-1} \bar{\boldsymbol{\eta}}_t^{\text{variance}}.$$

The remaining efforts are to characterize $\mathbf{B}_t^{\text{B}}$ and $\mathbf{C}_t^{\text{B}}$ for a batch size $B$. For the bias part, we have

$$
\begin{aligned}
\mathbf{B}_t^{\text{B}} &= \left(\mathcal{I} - \gamma \mathcal{T}^{\text{B}}\right) \circ \mathbf{B}_{t-1}^{\text{B}} \\
&= \left(\mathcal{I} - \gamma \tilde{\mathcal{T}}\right) \circ \mathbf{B}_{t-1}^{\text{B}} + \gamma^2 \left(\mathcal{M}^{\text{B}} - \tilde{\mathcal{M}}\right) \circ \mathbf{B}_{t-1}^{\text{B}} \\
&= \left(\mathcal{I} - \gamma \tilde{\mathcal{T}}\right) \circ \mathbf{B}_{t-1}^{\text{B}} + \frac{2\gamma^2}{B} \text{tr}\left(\mathbf{H}\mathbf{B}_{t-1}^{\text{B}}\right) \mathbf{H}, \quad t = 1, \ldots, n-1.
\end{aligned}
$$

For the variance part, we have

$$
\begin{aligned}
\mathbf{C}_t^{\text{B}} &= \left(\mathcal{I} - \gamma \mathcal{T}^{\text{B}}\right) \circ \mathbf{C}_{t-1}^{\text{B}} + \frac{\gamma^2 \sigma^2}{B} \mathbf{H} \\
&= \left(\mathcal{I} - \gamma \tilde{\mathcal{T}}\right) \circ \mathbf{C}_{t-1}^{\text{B}} + \gamma^2 \left(\mathcal{M}^{\text{B}} - \tilde{\mathcal{M}}\right) \circ \mathbf{C}_{t-1} + \frac{\gamma^2 \sigma^2}{B} \mathbf{H} \\
&= \left(\mathcal{I} - \gamma \tilde{\mathcal{T}}\right) \circ \mathbf{C}_{t-1}^{\text{B}} + \frac{2\gamma^2}{B} \text{tr}\left(\mathbf{H}\mathbf{C}_{t-1}^{\text{B}}\right) \mathbf{H} + \frac{\gamma^2 \sigma^2}{B} \mathbf{H}, \quad t = 1, \ldots, n-1.
\end{aligned}
$$

To obtain an upper bound on excess risk, we replace $\alpha$ in Assumption 2.2 of Zou et al. (2023) with $2/B$, the number of steps with $n := D/B$, and the noise level $\sigma^2$ with $\sigma^2/B$, then apply the proof of Theorem 2.1. Similarly, for a lower bound on excess risk, we replace $\beta$ in Assumption 2.4 with $2/B$, the number of steps with $n := T/B$, and the noise level $\sigma^2$ with $\sigma^2/B$, and apply the proof of Theorem 2.2. By doing the above, we obtain the following matching up to constant factors upper and lower bounds on the excess risk for mini-batch SGD:

$$
\begin{aligned}
\mathbb{E}\mathcal{R}(\bar{\mathbf{w}}) - \min \mathcal{R}(\cdot) &\eqsim \left(\frac{1}{n\gamma}\right)^2 \|\mathbf{w}_0 - \mathbf{w}^*\|_{\mathbf{H}_{0:k^*}^{-1}}^2 + \|\mathbf{w}_0 - \mathbf{w}^*\|_{\mathbf{H}_{k^*:\infty}}^2 \\
&\quad + \frac{1/B\left(\|\mathbf{w}_0 - \mathbf{w}^*\|_{\mathbf{I}_{0:k^*}}^2 + n\gamma \|\mathbf{w}_0 - \mathbf{w}^*\|_{\mathbf{H}_{k^*:\infty}}^2\right)}{n\gamma} \cdot \frac{k^* + (n\gamma)^2 \sum_{i>k^*} \lambda_i^2}{n} \\
&\quad + \frac{\sigma^2}{B} \cdot \frac{k^* + (n\gamma)^2 \sum_{i>k^*} \lambda_i^2}{n},
\end{aligned}
$$

where $k^* = \max\{k : \lambda_k \geq 1/(n\gamma)\}$, and a sufficient stepsize condition (see Lemma 4.1, Theorems 2.1 and 2.2 in Zou et al. (2023)) is

$$0 < \gamma \lesssim \min\left\{\frac{1}{\alpha \text{tr}(\mathbf{H})}, \frac{1}{\|\mathbf{H}\|_2}\right\} \eqsim \min\left\{\frac{B}{\text{tr}(\mathbf{H})}, \frac{1}{\|\mathbf{H}\|_2}\right\}.$$

The assumption $\|\mathbf{w}_0 - \mathbf{w}^*\|_{\mathbf{H}}^2 \lesssim \sigma^2$ implies

$$\frac{\|\mathbf{w}_0 - \mathbf{w}^*\|_{\mathbf{I}_{0:k^*}}^2 + n\gamma \|\mathbf{w}_0 - \mathbf{w}^*\|_{\mathbf{H}_{k^*:\infty}}^2}{n\gamma} \leq \|\mathbf{w}_0 - \mathbf{w}^*\|_{\mathbf{H}}^2 \lesssim \sigma^2,$$

which further simplifies the excess risk bounds to

$$\mathbb{E}\mathcal{R}(\bar{\mathbf{w}}) - \min \mathcal{R}(\cdot) \eqsim \left(\frac{1}{n\gamma}\right)^2 \|\mathbf{w}_0 - \mathbf{w}^*\|_{\mathbf{H}_{0:k^*}^{-1}}^2 + \|\mathbf{w}_0 - \mathbf{w}^*\|_{\mathbf{H}_{k^*:\infty}}^2 + \frac{\sigma^2}{B} \cdot \frac{k^* + (n\gamma)^2 \sum_{i>k^*} \lambda_i^2}{n}.$$

Finally, replacing $n = D/B$ in the bounds completes our proof. $\qquad\square$

*Proof of Corollary 2.* By $\lambda_i \asymp i^{-a}$, we can solve for $k^*$ to obtain $k^* \asymp (D\gamma/B)^{1/a}$. We then calculate the expected excess risk by Theorem 3 using the capacity and source conditions:

$$\mathbb{E}\mathcal{R}(\bar{\mathbf{w}}) - \sigma^2 \asymp \mathbb{E}\left(\left(\frac{B}{D\gamma}\right)^2 \|\mathbf{w}^*\|^2_{\mathbf{H}^{-1}_{0:k^*}} + \|\mathbf{w}^*\|^2_{\mathbf{H}_{k^*:\infty}}\right) + \frac{k^* + (D\gamma/B)^2 \sum_{i>k^*} \lambda_i^2}{D}$$

$$\asymp \left(\frac{B}{D\gamma}\right)^2 \sum_{i \le k^*} i^{-b+2a} + \sum_{i>k^*} i^{-b} + \frac{1}{D}\left(k^* + \left(\frac{D\gamma}{B}\right)^2 \sum_{i>k^*} i^{-2a}\right)$$

$$\asymp \left(\frac{B}{D\gamma}\right)^2 \max\left\{(k^*)^{1-b+2a}, 1\right\} + (k^*)^{1-b} + \frac{1}{D}\left(k^* + \left(\frac{D\gamma}{B}\right)^2 (k^*)^{1-2a}\right)$$

$$\asymp \max\left\{\left(\frac{D\gamma}{B}\right)^{(1-b)/a}, \left(\frac{D\gamma}{B}\right)^{-2}\right\} + \frac{1}{D}\left(\frac{D\gamma}{B}\right)^{1/a}.$$

We then discuss three cases.

1. When $b \le a$, we have

$$\mathbb{E}\mathcal{R}(\bar{\mathbf{w}}) - \sigma^2 \asymp \left(\frac{D\gamma}{B}\right)^{(1-b)/a} + \frac{1}{D}\left(\frac{D\gamma}{B}\right)^{1/a} \asymp \left(\frac{D\gamma}{B}\right)^{(1-b)/a},$$

   where the last equality is because $\gamma/B \lesssim 1$ so the first term dominates the second term. So the optimal hyper-parameters are $\gamma^* \asymp 1$ and $B^* = 1$.

2. When $a < b < 2a + 1$, we have

$$\mathbb{E}\mathcal{R}(\bar{\mathbf{w}}) - \sigma^2 \asymp \left(\frac{D\gamma}{B}\right)^{(1-b)/a} + \frac{1}{D}\left(\frac{D\gamma}{B}\right)^{1/a},$$

   so the optimal hyper-parameters are

$$0 < \gamma^* \lesssim 1, \quad 1 \le B^* \le D, \quad \gamma^*/B^* \asymp D^{a/b-1}.$$

3. When $b > 2a + 1$, we have

$$\mathbb{E}\mathcal{R}(\bar{\mathbf{w}}) - \sigma^2 \asymp \left(\frac{D\gamma}{B}\right)^{-2} + \frac{1}{D}\left(\frac{D\gamma}{B}\right)^{1/a},$$

   so the optimal hyper-parameters are

$$0 < \gamma^* \lesssim 1, \quad 1 \le B^* \le D, \quad \gamma^*/B^* \asymp D^{a/(2a+1)-1}.$$

Combining the second and third cases completes the proof. □

