# OpenReview forum: "How Does Critical Batch Size Scale in Pre-training?"
_ICLR.cc/2025/Conference — ICLR 2025 Poster_

### Official Review · Reviewer_A37b · 2024-11-03

**Soundness:** 4
**Presentation:** 4
**Contribution:** 4
**Rating:** 8
**Confidence:** 4

**Summary:**

The paper studies how a compute-optimal batch size, the critical batch
size (CBS) is influenced by scaling the amount of data or model
parameters in a Transformer-based language modeling objective. Most
importantly, the authors show a correlation between the training
duration and CBS, both empirically and theoretically. Importantly, the
experiments also show that model size does not have an effect on CBS.
Finally, a secondary but interesting part of the experiments concerns
allowing to change the training duration mid-training. It is shown
that a simple constant schedule combined with exponential weight
averaging at the end of training outperforms both "standard" cosine
decay and the recently proposed warmup-stable-decay (WSD) schedule.

The empirical study also includes a controlled setting to study the
effect of width- vs. depth-scaling on the CBS (the finding being that
both scaling methods influence the CBS similarly). Finally, the
authors additionally underline their empirical findings using
NTK/tensor programs and Gaussian linear regression theory.

**Strengths:**

For their experiments, the authors used C4, a de-facto standard NLP
research dataset, which makes interpretation and reproduction of the
paper of the paper easier.

A particularly strong section is that of relevant work, in which
various important related papers are summarized well.

I also really enjoy the "takeaway" boxes as section summaries.

**Weaknesses:**

The paper heavily relies on Chinchilla scaling theory and
Transformer-decoder training with the language modeling objective. It
does not account for settings which use fewer training steps than
Chinchilla-optimal training and does not study other data sets or
models. This means it is difficult to extrapolate the results to the
general domain of deep learning.

I would personally have preferred use of a more standard "dynamic
horizon" schedule such as WSD for the experiments in order to improve
interpretability/comparability with regard to other research. That
said, the finding regarding constant+EWA quality is important.

Weight decay was disabled for the experiments, which is hardly done in
practice.

**Questions:**

### Please address

In Proof of Theorem 1 (page 8, line 418-419), it is stated that "the
trajectory of the network approaches a limit as width tends to ∞". I
would wish for what exactly is meant by "trajectory" to be defined
more clearly. E.g., is it the trajectory of the weights during
training? Also, please cite a source for this statement.

Appendix D, Evaluation data size and frequency (page 18, line 964ff):
Which which model after what training have these evaluation variance
numbers been obtained?

### Minor comments

Page 3, line 158:
Please describe the meaning of θ.

Page 8, line 416:
$R(M, t)$ denotes the loss of network $N$ at time $t$.
-> $R(M, t)$ denotes the loss of network $M$ at time $t$.

In the experiments, ideally, the optimizer's ε hyperparameter would
also have been scaled with the model size
(https://arxiv.org/abs/2309.14322).

---

> ### Author Response · Authors · 2024-11-15
> **Response**
>
> Thank you for your constructive feedback. We have addressed each of your points as follows:
>
> > The paper heavily relies on Chinchilla scaling theory
>
> Please note that we do consider under-training and over-training compared to Chinchilla compute-optimal regimes (fig. 1 middle and fig. 6). This enables a more controlled comparison as we control for model size and data size separately, which further strengthens our claim.
>
>  >  "dynamic horizon" schedule
>
> Thank you for the suggestion. We do compare extensively using Sec 2.1, Appendix A. Especially our results of fig. 10 show that tuning the EWA decay rate high is needed for training longer. This plays a similar role in tuning down the learning rate when training longer, as suggested in [1].
>
>
> [1] Scaling Optimal LR Across Token Horizons Johan Bjorck, Alon Benhaim, Vishrav Chaudhary, Furu Wei, Xia Song.
>
> > Weight decay was disabled for the experiments.
>
> We disabled weight decay as we found it did not change the critical batch size in preliminary experiments for our setup. If this is crucial for reviewers’ judgment of our work we would be happy to redo some of the experiments with 1e-4 decoupled weight decay (as suggested in Wortsman 2023).
>
> We also note that our setup is based on the Olmo codebase. A previous work [2] (Figure. 5) also conducted experiments on the Olmo codebase and only observed very small difference for different weight decay values $0, 10^{-6}, 10^{-5}, 10^{-4}, 10^{-3}, 10^{-2}$ for autoregressive LM pre-training on C4.
>
> [2] Deconstructing What Makes a Good Optimizer for Language Models. Rosie Zhao, Depen Morwani, David Brandfonbrener, Nikhil Vyas, Sham Kakade
>
>
> > "trajectory" to be defined more clearly. e.g., is it the trajectory of the weights during training? Also, please cite a source for this statement.
>
> Thanks for the comment. Yes, we mean the trajectory of the weights. We will cite the Master Theorem of Yang and Hu (2021) in the proof as well.
>
> > Which model after what training have these evaluation variance numbers been obtained?
>
> As suggested in Sec 2.1, we use 151M models for most ablation experiments to decide hyper-parameters and evaluation configurations. We use C4 consistently throughout the paper as our training and evaluation data.
>
>
> Please note that the scaling of epsilon in Adam itself is an open research question and due to budget constraints we found that it has minor effects and didn’t extensively sweep over different other values. Note that we study models up to 1.2B, and [3] pointed out **For standard parameterization models with up to a billion parameters, the typical default value of 1e-8 is likely acceptable**
>
> [3] Scaling Exponents Across Parameterizations and Optimizers. Katie Everett, Lechao Xiao, Mitchell Wortsman, Alexander A. Alemi, Roman Novak, Peter J. Liu, Izzeddin Gur, Jascha Sohl-Dickstein, Leslie Pack Kaelbling, Jaehoon Lee, Jeffrey Pennington.
>
> > Typos on R(M,t) and \theta
>
> Thank you for pointing these out. We fixed them and updated the draft.

---

### Official Review · Reviewer_SERg · 2024-11-03

**Soundness:** 3
**Presentation:** 3
**Contribution:** 3
**Rating:** 6
**Confidence:** 4

**Summary:**

The authors reassess the notion of critical batch size (CBS) in the context of language model training, and investigate the impact of scale on CBS. They find that the CBS does not scale with model size in majority, but with data size, and highlight roles of optimizer choices, where they find exponential averaging with a constant LR to be performant, matching or outperforming cosine or WSD.

**Strengths:**

First of all, I thank the authors for looking into this problem — I believe the question of CBS is both extremely relevant and understudied. The paper is written in a clear manner, the experiments are very extensive, and the authors additionally provide theoretical studies. The work could be a starting point for further studies, e.g. going beyond Chinchilla optimal points.

**Weaknesses:**

While I very much appreciate the topic and investigation of the paper, I unfortunately have to be very critical of the experimental evaluation.

- First, the authors note in the Appendix that they disable weight decay of AdamW, without justification; this means effectively only studying Adam and not AdamW. However, weight decay is not only a major part of modern large scale training (see e.g. Chinchilla Fig A7 https://arxiv.org/pdf/2203.15556 or Wortsman et al. (2024) https://openreview.net/pdf?id=d8w0pmvXbZ), but it also strongly changes the training trajectory, where disabling it leads to a practical decrease of the effective learning rate even for a constant schedule; see e.g. Kosson et al. https://arxiv.org/pdf/2305.17212. Therefore, it is unclear how the findings would generalize to actual practical settings (proper baselines), and assuming the same results is misleading.
- The authors find a constant LR + EWA to be extremely performant. I have tried to replicate their results with the same hyperparameters (including disabled WD) in the same setting of LLM pretraining and fail to obtain the same results. While EWA can give a slight boost, it is far from the cooldown loss; even more, using e.g. a decay of 0.99 or higher results in a similar curve to the one of 0.9999 in Fig. 8 (increase in loss, but much earlier in training) and then EWA is *much worse* than the original model. Since the authors have not provided code, it is unclear to me where this discrepancy is coming from — I repeat this point as a question below.
- For most experiments, the schedule free optimizer performs surprisingly bad; if well tuned, I would think it should be at least as good or below the stable phase of WSD from my experience.

The combination of these points makes me concerned of some of the main empirical takeaways of the paper. To be clear, I believe the paper has much merit, and I do not want to reduce it to these concerns. However, I think addressing them requires either rerunning most experiments or major rewrites.

-- update:

The authors have engaged actively in the rebuttal and addressed the main concerns that I've raised. Expecting these points (weight-decay, fixed baselines, annealing to zero) to be well discussed in the camera-ready, and the findings of constant+EWA to be contextualized for the scope of the study, I have raised my score and look forward to the final version of the paper.

**Questions:**

I mention the main points in the section above, but repeat them here as questions (with some others):

- Why did the authors choose to disable weight decay?
- How did you use EWA in detail, or is there a specific implementation trick that was required?
- Did you tune SFO with the same sweeps as Adam?
- Why are the warmup steps chosen between 0.15, 0.25 and 0.35? These are quite large fractions, whereas in practice the warmup is often a very minor part of training (e.g. less than 5%).

I would be very happy to engage in a discussion about these points. More broadly, I hope the authors see my comments as not only critical but encouraging.

Another minor comment: I think the choice of colors for the plots is not ideal, as the contrast is very low and it makes it hard to read and compare lines (especially for people with color vision deficiency).

---

> ### Author Response · Authors · 2024-11-15
> **Response**
>
> Thank you for your constructive feedback. We have addressed each of your points as follows:
>
> > Schedule free optimizer
>
> Please note that From Fig. 2, schedule-free optimizer is competitive for small batch sizes, this aligns with what the original paper found. Notably the original paper only consider a batch size of around 2^9. However, it’s worse for large batch sizes even if we tuned the beta1 from 0.9, 0.95, 0.98.
>
> > Reproduction on constant LR + EWA.
>
> Please note that a very large EWA decay rate would cause instability but may lead to better convergence as suggested in the plot you referred to. As our goal is to reach a certain target loss, we use different EWA values for better convergence during the last part of the training.
> We included our implementation in the supplementary materials. Note that the performance gap also depends greatly on the setting and especially on training durations ana batch size.
>
> > Weight decay was disabled for the experiments.
>
> We disabled weight decay as we found it did not change the critical batch size in preliminary experiments for our setup. If this is crucial for reviewers’ judgment of our work we would be happy to redo some of the experiments with 1e-4 decoupled weight decay (as suggested in Wortsman 2023).
>
> We also note that our setup is based on the Olmo codebase. A previous work [1] (Figure 5) also conducted experiments on the Olmo codebase and only observed very small difference for different weight decay values $0, 10^{-6}, 10^{-5}, 10^{-4}, 10^{-3}, 10^{-2}$ for autoregressive LM pre-training on C4.
>
> Please note that the Wortsman et al. (2024) paper you mentioned shows that it does not affect the optimal LR for large enough models with size more than 85M (Figure E.10). To understand whether disabling weight decay would need a smaller learning rate, we add new experiments to show that our default value 3.16e-3 is consistently the best if the EWA decay rate is well tuned in Fig. 10.
>
> [1] Deconstructing What Makes a Good Optimizer for Language Models. Rosie Zhao, Depen Morwani, David Brandfonbrener, Nikhil Vyas, Sham Kakade
>
>
> > How did you use EWA and reproduction
>
> Please note that EWA is introduced in Sec 2.1 and we included all the possible values in Tab.3, as well as additional ablation studies in Fig. 10. Note that it’s common practice to tune EWA gradually from small values to larger ones. Especially we found that similar to the learning rate, the effectiveness of EWA+constant in different decay rates depends on the training duration.
> Please refer to our supplementary materials for reproduction.
>
> > Did you tune SFO with the same sweeps as Adam?
>
> Can you please clarify what do you mean by SFO? We didn’t use this abbreviation in our paper.
>
> > Why are the warmup steps chosen between 0.15, 0.25 and 0.35?
>
> Please note that choosing warmup proportional to data size is a common strategy that has been justified in previous work. In [2], they set the warmup period to be the minimum of the model size N and 20% of the total token budget. In the table 5 of [2], they also show that Durations of up to 4N achieve very similar and slightly better results. Therefore, we believe sweeping over 0.15, 0.25, 0.35 would be a reasonable design decision to achieve reasonable performance.
>
> [2] Resolving Discrepancies in Compute-Optimal Scaling of Language Models Tomer Porian, Mitchell Wortsman, Jenia Jitsev, Ludwig Schmidt, Yair Carmon.
>
> > Suggestion on colors.
>
> Thank you and we choose colors to reflect the ablations on different scales like model size and data size. We use Fig. 6 for side-by-side comparison to further support the claim.

---

> > ### Comment · Reviewer_SERg · 2024-11-20
> >
> > Thanks a lot for your reply and justifications! I will elaborate in a bit more detail below.
> >
> > > SFO
> >
> > SFO stands for Schedule-Free Optimizer, sorry that that wasn’t clear.
> >
> > > Warmup
> >
> > Thanks, I’m aware that the common setup is to use a warmup proportional to data size. My question was only towards the long length of the warmup, but yes, the justification is okay.
> >
> > > Weight Decay & EMA
> > >
> >
> > I do believe that the inclusion of weight decay in experiments is crucial. WD is very well accepted (or even standard) in the community for LLM training because it improves performance (e.g., already shown in Figure A7 in Chinchilla, also on C4) and its impact has been studied in different works. I cannot tell why the findings in the reference of Zhao et al. would differ to that (and it’s hard to tell from that Figure). Also, yes, the reference of Wortsman et al. shows that the optimal LR did not change for the values of WD because it is independent weight decay (side note: your code still seems to use the coupled one, referring to `p.mul_(1 - group['lr'] * group['weight_decay'])` in optim.py); I could see it not affecting the CBS, as you’ve noted, but I’m not sure. However, there is a noticeable boost by using WD, and this boost grows with model size (see e.g. Fig. 3 in their paper).
> >
> > Moreover, WD changes the learning dynamics and will have a larger impact on longer training runs and larger scale (e.g., when training is dominated by the ‘equilibrium / steady state phase’, Kosson et al. (2024) https://arxiv.org/pdf/2305.17212, and WD has a different effective LR, D’Angelo et al. (2024) https://arxiv.org/pdf/2310.04415). In particular, this notion of effective LR is important since you consider different LR schedules. For instance, a constant LR would usually first look worse with WD than without, and only upon cooldown (=WSD) the WD model will outperform the one without WD.
> >
> > A similar argument of WD also holds for the Schedule-Free Optimizer. Taken from their paper:
> >
> > > Weight decay for Schedule-Free methods can be computed at either the y or z sequences. We used decay at y for our experiments, as this matches the interpretation of weight-decay as the use of an additional L2-regularizer term in the loss. We found that computing the regularization at y gives significantly better performance on some problems including ImageNet and NanoGPT training.
> >
> > In a way, I therefore think that the baselines are not properly tuned and the claims (as other reviewers have noted) are too bold. Especially the takeaways in the main text such as *EWA consistently improves model training efficiency* are very strong (and I have doubts for when WD is enabled, for example that EMA would outperform a properly tuned cosine — the code seems to decay only to 10% per default?); even more so, considering the fact that you yourself emphasize that the optimal decay rate depends on the training duration in the Appendix. Also, do I understand right that you sometimes switch the EMA decay values over the course of training? This seems quite heuristic and hard to control for.
> >
> > I hate to be too critical, but I hope you see these points.

---

> ### Author Response · Authors · 2024-11-22
> **Further Response and Results**
>
> Thank you for your detailed feedback.
>
>
> > Hyper-parameter tuning.
>
>
> Please note that as indicated in Appendix, the baselines are well-tuned as we swept over learning rate [1e-3, 3.16e-3, 1e-2] for all schedulers and decay ratio [0.1, 0.2, 0.3] for WSD. For cosine scheduling, we first use optimal steps we found the relaunch that number of optimization steps as the maximum steps to make sure it would reach the the target loss right near the end of the training.
>
> > The impact of weight decay.
>
> We include the following experiments using decoupled weight decay for all the runs. We have hyper-parameter sweeps for 151M models using three schedulers - EWA, WSD, and ScheduleFree and our Constant+EWA. (due to time constraints we only swept over 4 batch sizes).
>
> lr - [3.16e-3, 1e-2], decoupled weight decay rate - [3e-04, 1e-04], batch size 512~4096, WSD decay ratio [0.1, 0.2, 0.3]. For the rest of hyper-parameters, we chose the best one from previous runs. We also sweep over the total training steps for both WSD and cosine to make sure they are close to optimal performance:
>
>
> | Batch Size | Training Duration (steps)    |
> |------------|-------------------------------|
> | 512        | 11000, 13000, 16000          |
> | 1024       | 8000, 8400, 10500                   |
> | 2048       | 5800, 6400, 9500                   |
> | 4096       | 3000, 3400, 3700             |
>
>
> The results are in an anonymous link at https://docs.google.com/document/d/1edDHtur9019FJDqLJ0hILIZiKuV4YXsEaMf4gzTlkbg/edit?usp=sharing.
>
> EWA+constant remains to be competitive. We see in the top figure that weight decay does slightly improve the schedule-free optimizer but not for others. We do used the original implementation of Schedule-free Optimizer provided in their Github, which applies weight decay at y as you suggested. All the schedulers remain nearly the same compared to the bottom figure reported in our paper. As our focus is the critical batch size, we additionally emphasize that adding weight decay does not (qualitatively) change the critical batch size too much.
>
> If the reviewers believe the claim should be revised, we are happy to do so. For instance, we propose that “EWA consistently performs as a competitive choice compared to the other three schedulers we evaluated, within the specific settings of our study.”
>
>
> > EMA decay values.
>
>
> We do not switch the value over the training but we show the values we swept in Tab. 3. We emphasize that as the learning rate is constant, we can more easily resume from checkpoints to reach certain target validation loss, which is well-suited for our case.

---

> ### Comment · Reviewer_SERg · 2024-11-23
>
> Thanks for the reply and the new experiments. Just to be clear: I did not imply that the hyperparameters that you have weren't properly swept or tuned, but you chose specific settings which -- under all current understanding in field -- lead to suboptimally trained models which you then compare to. I just wish the hyperparameter sweeps were done with a different setup in the first place.
>
> Moreover, to be very rigorous: when you say you use 'decoupled' weight decay, did you fix the line of code that I mentioned before? The attached code does not perform the original decoupled (or independent) weight decay which therefore would require much higher WD values such as 0.1. Values in the range of 1e-04 would not make much difference.
>
> Moreover, could you state the final learning rate for both WSD and cosine? As mentioned above, the code only decays to 10% per default and I do not see this config changed anywhere. This would lead to suboptimal final loss values as the LR is not fully annealed, especially for the large LR values.

---

> ### Author Response · Authors · 2024-11-24
>
> > I did not imply that the hyperparameters that you have weren't properly swept or tuned, but you chose specific settings which -- under all current understanding in field -- lead to suboptimally trained models which you then compare to. I just wish the hyperparameter sweeps were done with a different setup in the first place.
>
> It’s possible that weight decay can slightly increase the performance in pre-training. But the performance gain is minor as and Wortsman et al 2023, Zhao et al 2024 shows. Especially our primary focus is on critical batch size, rather than the optimal performance metric commonly emphasized in previous works. Thanks for the careful read and we can make some more discussions on this.
>
> > Moreover, to be very rigorous: when you say you use 'decoupled' weight decay, did you fix the line of code that I mentioned before? The attached code does not perform the original decoupled (or independent) weight decay which therefore would require much higher WD values such as 0.1. Values in the range of 1e-04 would not make much difference.  Final learning rate for both WSD and cosine? As mentioned above, the code only decays to 10% per default and I do not see this config changed anywhere. This would lead to suboptimal final loss values as the LR is not fully annealed, especially for the large LR values.
>
> We meant the default weight decay implementation in torch. Following the setup you described and using the same settings above, we swept over
>
> WD rate 1e-2, 3.16e-2, 1e-1 for LR 1e-2
>
> WD rate 3.16e-2, 1e-1, 3.16e-1 for LR 3.16e-3
>
> for all the schedulers. We decay cosine and WSD to zero as you suggested.
>
> Results are included in the doc (top one):
> https://docs.google.com/document/d/1edDHtur9019FJDqLJ0hILIZiKuV4YXsEaMf4gzTlkbg/edit?usp=sharing.
>
> We find that a large weight decay + decay to zero slightly improves cosine and schedule-free, but the performance between constan+EWA and cosine are close.
>
> Please also note that decay to 0 may improve loss but it may not be usually adopted. For example, [2] observe that *However, when evaluating on downstream benchmarks (see Section B.5), we see that annealing to zero can hurt metrics; we posit this comes from too early saturation.*
>
> [2] Scaling Laws and Compute-Optimal Training Beyond Fixed Training Durations
>
> Finally, we want to emphasize our focus and main contributions are on the empirical scaling behaviors of critical batch size followed by theoretical justifications. We did not claim in the main text that our constant+EWA approach offers a significantly improved convergence rate compared to other baselines. Instead, we highlighted its suitability for our study, treating it as a preparatory sub-section. The takeaway box is meant to be concise, where the claim “EWA consistently improves model training efficiency” is in comparison to pure “constant without EWA” based on our observations. We understand the takeaway as presented is a bit oversimplistic and would be happy to add more context. Thanks for noticing this and helping to properly focus our paper.

---

> > ### Comment · Reviewer_SERg · 2024-11-25
> >
> > Thanks a lot for the additional experiments! I am very well aware the focus of the paper was never on LR schedules or optimizer settings, but CBS. Nonetheless, its core is the experimental evaluation, which fundamentally depends on the former; so I felt compelled to point out flaws that could be important. In general, a paper on pretraining should use the currently known optimal recipes to train LLMs. There is also a point that I disagree with you: the increase in performance with weight decay is not so 'slightly' -- the differences might not look so significant on plots such as the ones in Wortsman et al., but going from say 3.30 to 3.25 is quite significant for LLMs, and even widens when you train bigger models for longer. It can literally save thousands of steps.
> >
> > For completeness, could you maybe provide the learning curves similar to Fig. 9?
> >
> > Re: annealing to zero, I agree it's usually not adopted (for cosine), but that's mostly related to potential continued training of the model. The provided paper showed the interesting point for downstream evals, yet it's quite an open question (cf their use of 'can'). In any case, the annealing should be done to low values (say 1e-5 - 1e-7) for proper loss values. This is actually the case for released large models that decay to 'only' 10% (since the max LR is much lower already).
> >
> > More broadly, I hope you actually somewhat agree with the points I raised to improve the paper and do not simply add these experiments for my satisfaction! I think this was quite a productive rebuttal. Conditioned on if you add all ablations into the paper (and replace the baselines with those that use tuned WD, fully decayed LR, update Figure 9 etc) and contextualize the findings properly, I will be happy to raise my score to 6 (and probably push the work above acceptance) :)

---

> > > ### Author Response · Authors · 2024-11-25
> > >
> > > Thank you for your thorough review, constructive feedback, and for raising the score! We have included the training curve in the previous link, which demonstrates that the schedulers are comparable even when both cosine and WSD converge to the target loss near the end of training.
> > >
> > > In our final version, we will incorporate additional results and discussions on scheduling strategies and the impact of weight decay. We appreciate you highlighting these aspects.

---

### Official Review · Reviewer_ivw4 · 2024-11-04

**Soundness:** 3
**Presentation:** 3
**Contribution:** 3
**Rating:** 6
**Confidence:** 4

**Summary:**

The paper revisits the role of batch size in scaling language models during pre-training. The core finding is that the critical batch size needs to scale alongside the model. However, both the critical batch size (CBS) and model scale (specifically, model width) show diminishing returns when scaled under a given training budget. The authors demonstrate and conclude that CBS scaling is dependent on data scale but invariant to model scale. Various hyperparameter ablations provide additional insights into relative pre-training performance efficiency.

**Strengths:**

1. The paper is well-written and largely easy to follow.
2. The related literature covers the important papers in the topic well.
3. Formalizations and hypotheses are clearly outlined and help understand results better.
4. Though personally I would like to reconsider its exact design and placement, the _Takeaway_ block was helpful while reading the paper first time.
5. The model scales reported in experiments are adequate in applying the insights to large-scale pre-training.
6. Formalizing the notion of critical batch size (CBS) through the 20% overhead assumption is novel and seemingly useful.
7. Considering various hyperparameters ablated across selected model scales is important and welcome in a scaling-related paper.

**Weaknesses:**

1. Some lower scale experiment with repetitions over different seeds to show the robustness of the findings (laws, exponents) and insights (data dependence and model scale invariance).
2. The work is mostly a benchmarking study with main contribution relying on the hypothesis constructed and how the experiment for it is setup, which therefore leaves more room for explaining some of the design choices, especially with model scale, hyperparameters (see, Questions below for examples).
3. Section 3.3 mentions how different hyperparameters were adjusted _to achieve optimal performance_ (at 302M), however, the grid search (for ablations) were done on a different model scale (151M), and thus is unclear what were the different HP settings considered in fitting points for the power law.
4. Theorem 1's proof by existence could do with more details and support, i.e., it is unclear how _training iterations t_ related to models of different width (thereby, scales), possibly different batch size scales and naturally different compute budgets under Chinchilla-optimality (see, Questions).
5. Despite leveraging theoretical insights from muP, there is only a limited grid search over learning rate values over the smallest available scale while that is apparently used as-is for even the larger model trainings, i.e., hyperparameters are not suitably scaled (this is unclear overall).
6. The outcome of the theoretical proofs and empirical results appear independent and do not knit the contributions as well.
    * Is there a way to connect the toy regression experiment with the scaling law exponent fits?


Minor issues/fixes:
1. Discrepancy of batch size 256 or 512 as the batch size for calculating overhead ratios
    * L52, Fig. 1,3,4,6,8 y-axes, L370, L888.
2. Should likely be `b > a` in L1299.
3. Typo on L366: perhaps should be "They" -> "Then"
4. Discrepancy in Fig. 5 caption where $C_{chin}$ comes to be around 6 as per the equation given for $B^*$.
5. Figure 4.a doesn't mention which model width or depth.
6. Figure 10 mentions _context length_ but has nothing to show for it.



Nitpicks:
1. Irregular and inconsistent plot sizes.
2. Recommended to refer sections/tables/figures in the Contributions list in Section 1.
3. Vertical lines in Table 4 separating different model sizes.
4. Parantheses in L1119-1122 would enable easier parsing for readers.

**Questions:**

1. Could the paper have bit more insight into why certain model scales were picked for certain experiments, since it is not always the smallest (151M) model that was selected?
2. Any reason explaining the worsening _efficiency_ for a 1.2B model on larger batch sizes in Figure 1.b) (right)?
3. For a study with even lesser bias or confounding factors, would it make sense to make conclusions on the critical batch size (CBS) or its scaling exponents without an EWA of model weights?
    * In a similar vein, can we see in Figure 2 what only constant LR schedules look like?
    * Is cosine here with or without warmups?
4. Why does Figure 2 (right) have different step lengths for the same batch size assuming similar compute budgets overall, given same model size?
    * Similar to above, why does Figure 9 all have different lengths?
5. Why do we expect the rankings of different schedulers found on 151M scale would transfer to larger scales? Do we have a literature or empirical reference for it?
6. Is the only notion of _efficiency_ in paper as denoted by the overhead ratio metric as defined by the measure on batch size 256/512?
    * For every result seen as plot/table, does it mean there was an equivalent run made on a batch size 256/512?
7. Would Figure 3 result hold for larger models with bigger context sizes? (I am not asking for an experiment at such scales but a cheaper experiment that could serve as proof-by-contradiction)
    * How does the model size change when Figure 3 writes about a model of size 151M but with context size ranging from 512-4096?
    * Are model sizes calculated without the embedding parameters?
8. Could the 20% and the resulting 5 in Section 3.2 be more generalized in the formalism?
    * Could the readers have a clearer reference/intuition for the 20% overhead?
9. Could L318-319 have more support?
10. Does Figure 5 have empirical backing regarding the predicted CBS and the formalism of staying within the 20% margin of steps for similar loss?
    * Or if the decoupled prediction for batch sizes hold _reliably_ for different scales or ablations such as architecture, datasets, hyperparameters?
11. How exactly are the tuning decisions undertaken especially in Section 3.3 (example, L371-372)?
12. In Theorem 1, is it expected or meant to be $w_2 > w_1$? If so, then there is a typo and makes a significant difference to the interpretation.
13. What is the significance of _t_ in Theorem 1? Should we compare losses under similar tokens for two vastly different model scales, especially following Chinchilla?
14. What is the specification or how is $\mathbf{H}$ defined in Section 4.2?
15. Is there a more intuitive summary for the proofs for Theorem 2 and 3 especially for readers (like me), not familiar with Zou et al. (2023)?
16. How exactly does the _hybrid evaluation_ (L968) provide _more accurate_ information?
17. What or how are the hyperparameters (HPs) for parameter sweeps ranked or ordered (L977-979)? Does a different ordering yield potentially different results?
    * What are the default values for other hyperparameters when tuning each of these in order?
    * Assuming in this order that LR is considered to be the most important HP, why tune the least important HPs (in $\beta_2$ and $\lambda$) for each model scale and not LR (Table 4)?
18. I may have missed it but could there be a reference for how the power law model fit is made for the values in the caption of Figure 7?


Overall, the paper reads nice and does touch on an important and often under-studied aspect in scaling literature.
However, the paper writing and presentation raises some eyebrows with really big plots, white spaces used, and a theory which feels like a sub-paper than something that brings the story together.
Given the expensive space of empirical experiments, the authors did well to consider multiple design choices and study them.
Unfortunately, that also opens up more questions on how these designs were arrived at and that clarity would be my primary criticism.
One other thing would be the lack of _a_ clear takeaway in terms of _how_ a pre-training practitioner could use the insights from this paper when scaling (model or data).


Scores may be increased depending on suitable responses to most of the points raised above.
Thank you for the paper, it was a nice read overall.


PS: I was unaware of Zou et al. 2023 and have not verified the proof on Pages 22-24.

---

> ### Author Response · Authors · 2024-11-15
> **Response [1]**
>
> Thank you for your constructive feedback. We have addressed each of your points as follows:
>
> > different seeds, design choices
>
> Thanks for the feedback. Due to computation constraints, we did not sweep over multiple seeds but focused mainly on different hyper-parameters. But from our variance measure experiments in Appendix D, our runs are pretty consistent across seeds.
> Please note that we have included many results in the Appendix to study the effects of major hyper-parameters.
>
> > hyper-parameters are done with small-scale 151M models
>
> Due to computation constraints, we use small-scale 151M models to study some hyper-parameters including context lengths, momentum $\beta_1$, scheduling and learning rate studies. However, for important ones that can lead to performance boost like EWA decay rate $\tau$ and $\beta_2$, we swept all the values as indicated in Tab.3.
>
> > A limited grid search over learning rate
>
> Note that we have swept learning rate for various model scales as suggested by Tab.3.  We found that under our constant+EWA strategy, learning rate does not affect the steps required to reach a target loss too much. In Appendix Fig. 10, we include more studies to understand the the impact of learning rate. We found that our default value 3.16e-3 is stable across the three regimes we considered. To improve model performance with a fixed learning rate, increasing the training duration may require a larger EWA decay rate for optimal results.
>
> > Connections between the outcome of the theoretical proofs and empirical results? Is there a way to connect the toy regression experiment with the scaling law exponent fits?
>
> Please note that our theoretical proofs corresponds to two controlled settings we consider: (we included the following into Intro Sec 1.1, Sec 1.2 as well)
> Empirically:
>
> 1) If we scale up training duration D while keeping N fixed (Figure 1, middle), the critical batch size increases to a similar degree.
>
> 2) However, we find that CBS remains nearly invariant when scaling up N while keeping D fixed (Figure 1, right), suggesting that CBS weakly depends on model size N but more strongly depends on data size D.
>
> Corresponding to the two cases above, theoretically:
>
> 1) In infinite width regimes [Yang et al, 2021], training dynamics and performance of the networks become effectively independent of the model size. Consequently, the critical batch size remains nearly invariant when scaling up the model size beyond this point, indicating that larger models do not require proportionally larger batch sizes to achieve optimal training efficiency.
>
> 2) Consider mini-batch SGD with $D$ samples in the least square problems under power-law source and capacity conditions. The CBS, which enables mini-batch SGD to achieve the minimal expected excess risk while ensuring the fastest possible serial runtime, is given by $B^*(D) = \Theta(D^c)$, where the exponent $c\ge 0$ is determined by the exponents of the source and capacity conditions. In the regime where the variance error tends to be dominant, we have $c>0$, indicating CBS grows with data size.
>
> > Discrepancy of batch size 256 or 512 as the batch size for calculating overhead ratios
>
> Please note that we use $B_{\text{opt}}=256$ to determine the target loss consistently, but report the relative number of steps with 512 batch size results as the denominator because we are only interested in large batch size regimes to build intuitions on CBS. We didn’t claim they are the same.
>
> > Should likely be b > a in L1299. Typo on L366: perhaps should be "They" -> "Then". Discrepancy in Fig. 5 caption $C_{chin}$. Figure 10 mentions context length. Plot sizes.
>
> Thanks for pointing these out. We have greatly revised the draft to fix those issues.
>
> > Figure 4.a doesn't mention which model width or depth.
>
> Please refer to Tab. 5 for the concrete setup, as mentioned in the main text.
>
> > Recommended to refer sections/tables/figures in the Contributions list in Section 1.
>
> Thank you for the suggestion. We added Sec 2.1 and 2.2 accordingly for better introduction of our contributions.
>
> > Vertical lines in Table 4 separating different model sizes. Parantheses in L1119-1122 would enable easier parsing for readers.
>
> Thank you for the suggestion and we would include the changes in the next draft.

---

> > ### Author Response · Authors · 2024-11-15
> > **Response [2]**
> >
> > > Any reason explaining the worsening efficiency for a 1.2B model on larger batch sizes in Figure 1.b) (right)?
> >
> > It’s possible that constraining the data to be only 3.07B tokens would make 1.2B models harder to train using batch size $2^{13}$.
> >
> > > Why do we expect the rankings of different schedulers found on 151M scale would transfer to larger scales? Do we have a literature or empirical reference for it?
> >
> > We do not claim that the results can transfer without proper hyper-parameter tuning. But we do consider a realistic scale, e.g. a batch size of millions of tokens, compared to previous works. Benchmarking schedulers helps justify the use of EWA can avoid setting training durations beforehand.
> >
> >
> > > Could the paper have bit more insight into why certain model scales were picked for certain experiments, since it is not always the smallest (151M) model that was selected?
> >
> > In experiments where model size is controlled, we use 302M models because that is the medium one so we can scale the data proportionally compared with compute-optimal training of 85M, 151M, 604M, and 1.2B models. This makes comparison more controlled and the model size is small enough for sweeping over hyper-parameters.
> > For Chinchilla settings and other settings where data size is controlled, we consider 5 different model sizes that can be scaled through certain modifications as shown in Tab. 2 due to computational constraints.
> > For understanding the impact of hyper-parameters, we use 151M models as they are more efficient to train.
> >
> > > For a study with even lesser bias or confounding factors, would it make sense to make conclusions on the critical batch size (CBS) or its scaling exponents without an EWA of model weights?
> > In a similar vein, can we see in Figure 2 what only constant LR schedules look like?
> > Is cosine here with or without warmups?
> >
> > We expect CBS to exist regardless of the learning rate schedule, as shown in previous work (Shallue et al., 2019; McCandlish et al., 2018). But as clarified in Sec 2.1, we hope to study CBS on the premise that other hyper-parameters are well-tuned. In other words, we don’t want to arrive at an unreliable conclusion about CBS if important hyper-parameters are very sub-optimal. Moreover, cosine scheduling is also run with warm-up steps.
> >
> >
> > > Why does Figure 2 (right) have different step lengths for the same batch size assuming similar compute budgets overall, given same model size?
> > Similar to above, why does Figure 9 all have different lengths?
> >
> > Please note that we are not recording the final performance at the end of training but just the number of steps to reach the target loss indicated by the dashed line. Only the best-performing curves are shown so the lengths can vary depending on the settings.
> >
> > > Is the only notion of efficiency in paper as denoted by the overhead ratio metric as defined by the measure on batch size 256/512?
> > For every result seen as plot/table, does it mean there was an equivalent run made on a batch size 256/512?
> >
> > Yes please refer to Fig. 13 for all the results by taking batch size 64 performance as the denominator. As shown in Fig.13, because B_{opt}=256 is in the linear scaling regime, we use it to determine the target loss throughout the paper.
> >
> > > Would Figure 3 result hold for larger models with bigger context sizes? (I am not asking for an experiment at such scales but a cheaper experiment that could serve as proof-by-contradiction)
> > How does the model size change when Figure 3 writes about a model of size 151M but with context size ranging from 512-4096?
> > Are model sizes calculated without the embedding parameters?
> >
> > As this is not our main focus of the paper but justifying the design decisions of using a fixed context length of 512 throughout the paper, we didn’t consider long-context models and larger models. Model sizes are calculated with non-embedding parameters.
> >
> > > Could the 20% and the resulting 5 in Section 3.2 be more generalized in the formalism?
> > Could the readers have a clearer reference/intuition for the 20% overhead?
> >
> > Yes the overhead level can be generalized according to specific practical settings and we found that these does not affect the fitted scaling laws too much. For example, when having the following 10%, 50% overhead, we get data scaling laws for 302M models of similar form as
> >
> > 10%: B* = 20.67 * D^(0.48)
> >
> > 20%: B* = 22.91 * D^(0.47)
> >
> > 50%: B* = 30.50 * D^(0.44)
> >
> > > Could L318-319 have more support?
> >
> > Sure, we will add reference to Fig. 1 left, which shows that for 1.2B models, up to 2^11 batch sizes, linear scaling regimes still hold.

---

> ### Author Response · Authors · 2024-11-15
> **Response [3]**
>
> > Does Figure 5 have empirical backing regarding the predicted CBS and the formalism of staying within the 20% margin of steps for similar loss?
>
> Or if the decoupled prediction for batch sizes hold reliably for different scales or ablations such as architecture, datasets, hyperparameters?
>
> Please note that in Sec 2.1 we have explained that we only take into account the best-performing run for each batch size. So the scaling laws hold if we have models well-tuned across hyper-parameters we consider in the paper. We only consider transformer-based auto-regressive language models on C4 as they are canonical settings that are practical and can be reproduced more easily.
>
> > How exactly are the tuning decisions undertaken especially in Section 3.3 (example, L371-372)?
>
> Please note that we have mentioned that we scale the warmup tokens proportionally: in Fig. 1 we have compute-optimal setting 1x, and the 0.28x, 0.5x for under-training, 2x, 4x for over-training. If we set the warmup tokens for compute-optimal setting as $0.25C_{Chin}N$, then we have $0.07C_{Chin}N$, $0.125C_{Chin}N$ for over-training and $0.5C_{Chin}N$, $C_{Chin}N$ for over-training.
>
> > In Theorem 1, is it expected or meant to be w2>w1? If so, then there is a typo and makes a significant difference to the interpretation
>
> No, only the two widths are greater than $w$, there is no relation between the two widths
>
> > What is the significance of t in Theorem 1? Should we compare losses under similar tokens for two vastly different model scales, especially following Chinchilla?
>
> Yes, this theorem specifically only holds for models with different widths trained for the same number of tokens (and thus does not hold for Chinchilla setup where tokens increase with increasing model scale). So $t$ refers to the fixed number of tokens.
>
> > What is the specification or how is H defined in Section 4.2?
>
> As shown in the beginning of Sec 4.2, H is the covariance matrix of the Gaussian data distribution.
>
> > Is there a more intuitive summary for the proofs for Theorem 2 and 3 especially for readers (like me), not familiar with Zou et al. (2023)?
>
> Please refer to Sec 1.2, our newly added section, for an overview of the theory:
>
> - Theoretically, maximal update parameterization suggests that, beyond a certain point, increasing the width of the neural network (while keeping data size fixed) does not further increase the critical batch size. In contrast, by analyzing a simple least-squares regression with mini-batch SGD, we provide a theoretical basis for how the critical batch size continues to scale with increasing data size.
> - In infinite width regimes, training dynamics and performance of the networks become effectively independent of the model size. Consequently, the critical batch size remains nearly invariant when scaling up the model size beyond this point, indicating that larger models do not require proportionally larger batch sizes to achieve optimal training efficiency.
>
> > How exactly does the hybrid evaluation (L968) provide more accurate information?
>
> The evaluation is for making sure the recorded #steps required to reach a goal loss would not be overly estimated because of infrequent evaluations. So we adopt the strategy to evaluate the runs more frequently when near the end of training.
>
> > What or how are the hyperparameters (HPs) for parameter sweeps ranked or ordered (L977-979)? Does a different ordering yield potentially different results?
> What are the default values for other hyperparameters when tuning each of these in order?
> Assuming in this order that LR is considered to be the most important HP, why tune the least important HPs (in β2 and λ) for each model scale and not LR (Table 4)?
>
> Note that we have swept learning rate for various model scales as suggested by Tab.3.  We include the search details for completeness but find that ordering does not result in too many differences. For example, our newly added experiments in Fig. 10 shows that our default learning rate 3.16e-3 is consistently the best among others with different EWA decay rates. We highlight the default values in Tab.3 using bold fonts.
>
> > I may have missed it but could there be a reference for how the power law model fit is made for the values in the caption of Figure 7?
>
> The scaling law details are included in Sec 3.2 and Sec 3.3. Appendix E also includes additional details and interpretations.
>
> > How these designs were arrived at.
>
> We include quite a few additional experiments on understanding design choices like learning rate in the Appendix. Please let us know which parts are unclear and we can clarify further.

---

> > ### Author Response · Authors · 2024-11-15
> > **Response [4]**
> >
> > > How a pre-training practitioner could use the insights from this paper when scaling (model or data)?
> >
> > We have include Sec 1.1 for clarifying the empirical takeaways. Especially, our empirical finding that CBS scales primarily with data size implies that when scaling up data, one can reduce serial training time through greater data parallelism due to the increase of CBS, without a loss in computational efficiency that can be measured by floating point operations (FLOPs).

---

> > > ### Comment · Reviewer_ivw4 · 2024-11-22
> > > **Response to Rebuttal [1/2]**
> > >
> > > Thank you for addressing many of the comments made. The updated paper does ease the reader much better and Figure 5 is a good addition.
> > >
> > > > constraining the data to be only 3.07B tokens would make 1.2B models harder to train using batch size 2^13
> > >
> > > Does this rank order change perhaps with higher EWA length-scale fraction?
> > > Given the fixed data size and Chinchilla training, I believe the trainings for all models >302M use no longer sub-epochs and tokens are repeated.
> > > For the decoupling argument, that makes sense but might be _more appropriate_ to try the same experiment(s) with a different dataset (> 20*N tokens) that keeps all models sub-epoch, thereby making it a fairer comparison across model sizes even for fixed data size.
> > >
> > > > Benchmarking schedulers helps justify the use of EWA can avoid setting training durations beforehand
> > >
> > > This likely follows from [1]? To clarify, do you suggest that instead of using LR cooldown towards the end of training budget, EWA is a better strategy?
> > >
> > > Is it possible to add *Constant+Cooldown* to Figure 2 (in updated draft) for completeness w.r.t [1].
> > > Adding this would certainly make the constant+EWA contribution more sound.
> > >
> > > > we hope to study CBS on the premise that other hyper-parameters are well-tuned
> > >
> > > How does this work out for larger scales when the tuning is feasible only on the smallest scale available of 151M?
> > >
> > >
> > > *References*:
> > >
> > > [1] Alexander Ha ̈gele, Elie Bakouch, Atli Kosson, Loubna Ben Allal, Leandro Von Werra, and Martin Jaggi. Scaling laws and compute-optimal training beyond fixed training durations, 2024

---

> > > ### Comment · Reviewer_ivw4 · 2024-11-22
> > > **Response to Rebuttal [2/2]**
> > >
> > > New questions:
> > >
> > > 1. Not sure what this means in caption of Table 3 (L1053): "Values in the parathesis mean used not for every sweep". Could the caption be made a bit more clear? Does bolding imply a hard selection and not top-1 from a grid sweep? Does not bolding any (EWA decay, momentum, etc.) imply a full sweep for each model scale?
> > >
> > > 2. Table 5 on HPs is still not completely transparent. Using 151M for sweeps is understandable. However, given the finite set of models reported, would be good to include HPs that were *used* for training each of them. At least for the Chinchilla setting. Especially given that for the decoupling experiments certain ad hoc tuning were applied for larger model scales and are not necessarily reproducible experiments as it appears.
> > >
> > > 3. In Fig. 13, may I ask how long did the training with batch size 2^6 take on the 1.2B model?
> > >
> > > General recommendations:
> > > 1. Mention model sizes in Figure captions/titles. It is sometimes hard to find the corresponding model size from the related text.

---

> ### Author Response · Authors · 2024-11-23
>
> >I believe the trainings for all models >302M use no longer sub-epochs and tokens are repeated.
>
> All the runs are one-epoch or sub-epoch without repeating data, a common practice in pertaining. C4 has 153.6 billion tokens and we are far from using all the data.
>
> >  To clarify, do you suggest that instead of using LR cooldown towards the end of training budget, EWA is a better strategy?
>
> We find that in our particular settings, constant+EWA is competitive. In contrast, [1] found that "We also experimented with an exponential moving average (EMA), which performed worse than SWA, and therefore we do not report EMA.".
> As indicated in the main text: as we focus on the number of training steps needed to achieve a target validation loss, learning rate decay strategies typically require predefining the total training duration. To address this, we propose using exponential weight averaging (EWA) to achieve the desired target validation loss, a simple approach that matches other popular choices (Figure 2). This enables training beyond fixed durations or data size, allowing to resume training from checkpoints until the target validation loss is achieved.
>
> > adding constant + cooldown.
>
> WSD is exactly the Constant + Cooldown you are referring to, which was released earlier according to the arxiv timestamp https://arxiv.org/abs/2404.06395.
>
> > How does this work out for larger scales when the tuning is feasible only on the smallest scale available of 151M?
>
> For important hyper-parameters, we already mentioned in Appendix that we swept for all Chinchilla settings for all model sizes using the ones in Tab. 3 and 4. The experiments where we only used 151M models including context length etc are not primary experiments. Those are already on a much larger scale than the papers we discussed.
>
> > Hyper-parameter captions and reproducibility
>
> Learning rate (3.16e-4), (1e-2) are the ones we experimented for 151M models and found that with EWA it does under-performs 3.16e-3. EWA decay rate (0.99995) is only for long 1.2B runs. We would clarify further in the latest draft.
>
> Bold font means the default hyper-parameters that can closely reproduce our results without extensive tuning. Not bolding implies a full sweep for each model scale. Please note that without such annotation makes it harder to reproduce as hyper-parameter search requires computational resources. For important ones like $\beta_2, \tau$ we listed separately in Tab.4. Please also refer to our actual code implementation where we keep yaml files separately for each experiment. We can also share wandb run reports upon request after peer review.
>
> > how long does it take for training batch size 64, model size 1.2B
>
> It takes 22h 12m 37s on a node with 8 80GiB A100s.
>
> > Mention model sizes in Figure captions/titles. It is sometimes hard to find the corresponding model size from the related text.
>
> We would incorporate the feedback into our latest draft.

---

> > ### Comment · Reviewer_ivw4 · 2024-11-23
> > **Response to Rebuttal**
> >
> > Thanks for the prompt response and for clarifying many more points.
> >
> > > As indicated in the main text: as we focus on the number of training steps needed to achieve a target validation loss, learning rate decay strategies typically require predefining the total training duration.
> >
> > Thanks for clarifying. Well agreed.

---

> ### Comment · Reviewer_ivw4 · 2024-11-23
> **Increasing score**
>
> Right now, the key points I take with me from your paper:
> * A formalism for critical batch size (CBS).
> * The existence of an increasing CBS under Chinchilla scaling.
> * Decoupled influence of the number of tokens seen on CBS for a fixed-size model.
> * Neat theoretical proof of a CBS limit with model-size scaling (along width) under a fixed size of tokens.
>
> Thus, the paper is an interesting read with clean empirical results to support the claims.
> A key practical aspect that emerges: when scaling up model size, the critical batch size should not be scaled unless accompanied by an increase in dataset size.
>
> I increase my score to 6.
> Thank you for the paper.

---

> > ### Author Response · Authors · 2024-11-23
> >
> > Thank you a lot for your prompt reply and increasing the score. Please let us know if you have additional questions so we can clarify further before the deadline.

---

### Official Review · Reviewer_76qC · 2024-11-06

**Soundness:** 3
**Presentation:** 3
**Contribution:** 3
**Rating:** 8
**Confidence:** 3

**Summary:**

The paper studies how the critical batch size (CBS) - the threshold beyond which increasing batch size causes diminishing return - scales with the model and data size for pre-training language models. The authors conduct careful experiments on models ranging from 82 million to 1.2 billion parameters and show that the CBS slightly scales with data size rather than the model size (remains invariant to model size). The authors also provide theoretical justifications for this phenomenon using infinite-dimensional least squares regression and the infinite-width limits of neural networks. These findings have important implications for designing efficient pre-training strategies.

**Strengths:**

- The paper is well-written with clear organization. This work would be a valuable contribution to the ICLR community. I especially appreciated the “key takeaways summary” after each section.
- The experimental design is rigorous; for example, decoupling various hyperparameters makes the claims more convincing.
- Detailed experimental procedures are provided in Appendix D.
- The formalization of CBS (beyond [1] “An empirical model of large-batch training”) would be helpful in the literature. The key findings that the CBS scales slightly with data size (and stays invariant to model size) are interesting. These insights have important implications for efficient pre-training strategies.

**Weaknesses:**

- The experimental scope is limited to models up to 1.2B parameters trained on C4, which may not fully capture scaling behaviors at larger scales (e.g., models with over 50B parameters). On a similar note, key ablation studies are primarily conducted on smaller models (with C4). However, given the careful experimental design and clear theoretical analysis, I do not believe that these impact the validity of the findings.
- It would be helpful to have a dedicated section discussing the limitations of both the theoretical analysis and empirical findings (e.g., Gaussian data distribution).
- However, I believe that the work has substantial limitations that would prevent me from recommending acceptance.

**Questions:**

- (Minor) It would be nice to use $\times$ instead of * (asterisk) in line 135.
- (Minor) The color contrast in Figure 9 can be improved.

---

> ### Author Response · Authors · 2024-11-15
> **Response**
>
> Thank you for your constructive feedback. We have addressed each of your points as follows:
>
> > The experimental scope is limited to models up to 1.2B parameters trained on C4. However, given the careful experimental design and clear theoretical analysis, I do not believe that these impact the validity of the findings.
>
> Thank you for the note. Due to computational constraints, we cannot extend our study to a larger scale. But we have carefully controlled for different factors in the settings we considered.
>
> > It would be helpful to have a dedicated section discussing the limitations of both the theoretical analysis and empirical findings (e.g., Gaussian data distribution).
>
> Thanks for the suggestion. Please note that we do include the discussion of our setting is limited to C4 data in the Conclusion section. We would include the following: The theoretical analysis assumes Gaussian data distribution and infinite-width neural networks, which may not hold for real-world, non-Gaussian, high-dimensional data or finite models, potentially affecting the generalizability of CBS scaling laws. Empirical findings are restricted by limited hyperparameter sweeps and computational resources, which may prevent a full exploration of optimal configurations, particularly across varied tasks and larger architectures.
>
> > It would be nice to use instead of * (asterisk) in line 135. The color contrast in Figure 9 can be improved.
>
> Thank you for the suggestions and we will include your feedback into the final version.

---

> > ### Comment · Reviewer_76qC · 2024-11-25
> >
> > Thank you for your reply. I acknowledge reading the author's response and other reviewers' comments. I will maintain my current score.

---

### Official Review · Reviewer_FgiN · 2024-11-06

**Soundness:** 3
**Presentation:** 3
**Contribution:** 3
**Rating:** 6
**Confidence:** 3

**Summary:**

The paper investigates the scaling behavior of critical batch size in the pre-training of autoregressive language models.
They first define the critical batch size (CBS) as the point where increasing the batch size no longer leads to significant gains in computational efficiency (>20% overhead when doubling the batch size vs. a linear scaling). They then perform experiments to determine the CBS of autoregressive Transformer-based language models of varying scales, finding that CBS scales primarily with the size of the training dataset rather than model size. They provide theoretical support for this finding by studying infinite-width limits of neural networks and infinite-dimensional least squares regression problems.

**Strengths:**

- The paper provides an interesting finding that the critical batch size scales mostly with data set size, and is largely invariant to model size. This is a relevant and, to my knowledge, novel insight.
- The paper considers models ranging from 85 million to 1.2 billion parameters and thus covers a reasonably large domain of models.
- I really liked the highlighted practical takeaway blocks throughout the paper, which made it easy to understand, well-structured, and accessible.

**Weaknesses:**

Some of the takeaways seem to me a bit too bold or not backed by enough evidence for the given claim.

- For example, in Section 2.2, they compare the efficiency of learning rate schedules across batch sizes by comparing the number of steps to achieve a given target validation loss. They conclude that "EWA consistently improves model training efficiency. [...] while outperforming Cosine for large batch sizes [...] and even with appropriate learning rate decay, [Cosine] underperforms our constant+EWA strategy in large-batch settings." (line 188). However, looking at Figure 2b), we can see that the training duration of cosine was chosen inefficiently. It continues to decay well beyond the target validation loss, achieving the target at roughly 50% of its decay schedule. It is also clear to see for the WSD schedule, which hits the target loss shortly after starting its decay, i.e. it is still halfway in its decay phase when crossing the threshold loss. I believe making a statement like "EWA consistently improves model training efficiency" requires a more rigorous empirical analysis. For instance, Schedule-Free [1] suggests a similar running average strategy to improve efficiency, providing a much more comprehensive and rigorous analysis.
- Similarly, Section 2.3 claims that different context lengths have similar CBS, based on a single (comparably small) model & dataset. It seems that the lines start to diverge a bit more for larger batch sizes, so is it possible that this is more pronounced for other settings (e.g. larger models, more training data, etc.)?
- All claims are made for the definition of CBS using 20% overhead. The authors state that "20% can be replaced by any other suitable measure of increase from linear scaling". I wonder how the results and conclusions would change if one varies this parameter, e.g. to 10% or 50%. For example, what would a plot look like of the CBS per model or data size as a function of the overhead?

[1] Aaron Defazio, Xingyu Alice Yang, Harsh Mehta, Konstantin Mishchenko, Ahmed Khaled, Ashok Cutkosky; "The Road Less Scheduled"; arXiv 2024; <https://arxiv.org/abs/2405.15682>

**Questions:**

- A central takeaway of the paper is that "CBS remains invariant when scaling up N [model size]" (line 93). However, isn't this partially contrasted by the results in Section 2.4, where you write "Figure 4 shows that increasing depth and width [and thus model size] can both slightly increase the CBS" (line 264)?
- I found the plots to sometimes be a bit confusing or less accessible. Some suggestions:
  - Could you add the linear scaling line as well as the 20% region?
  - Could you highlight the critical batch sizes, e.g. by a star marker?
  - Nit: The font sizes are inconsistent between subplots, e.g. the x-axis labels between Figure 1a and the subplots of Figure 1b.
  - In the legend of Figure 1b (left), what is the number in parenthesis? Is it the (relative) number of training samples?
  - The lines are sometimes hard to read. E.g. in Figure 2a, the Constant+EWA line is very bright. Similarly the lines in Figure 2b. I also had trouble distinguishing the shades of blue, e.g. in Figure 1. But it does look pretty :)
  - Nit: The white spaces in Figure 4 are a bit weird. There is only a small space between the subcaption of subplot (a) and much more white space between the subcaption (b) and its subplot.
- In line 110 you mention that "traditional learning rate decay strategies typically require predefining the total training duration" and you mention Defazio et al., (2024). However, isn't Schedule-Free a counter-example? Schedule-Free is also using a, at least to me, a relatively similar approach to your "constant+EWA" strategy, no (e.g. comparing the equation in line 158 to Eq. (5) by Defazio)? Could you elaborate on how your method differs from Schedule-Free?
- For most of the paper, you sweep the batch size between $2^6$ and $2^{13}$ with the exception of Figure 3, where the batch size is between $2^{16}$ and $2^{22}$. Is this because in this figure, you count the number of tokens, so practically batch size * tokens per batch?
- You provide a scaling law fit of the CBS. Did you try checking how close your thus predicted CBS tracks the true CBS for "unseen data points", i.e. test its predictive power?
- In the concluding remarks you write "Our findings contribute insights into [...] particularly highlighting the role of hyper-parameters such as learning rate scheduling and optimizer settings" (line 530). What "optimizer settings" are you referring to?
- Nits:
  - Line 120: I believe the "Refer" is superfluous.
  - Line 157: There is probably a closing parenthesis missing after the second citation.
  - Line 185: Typo for "maximum".
  - Figure 3a doesn't need to be a subplot as there is no other subplot.
  - Line 260: This sentence seems grammatically weird to me "As the main result in Figure 1 only involves a single way for scaling up models [...]". I might also have misunderstood the sentence. For example, doesn't scaling from 151M to 302M double the depth, and therefore Figure 1 involves multiple ways of scaling up models?
  - Line 271: "Compute-optimal" should probably be lowercase.
  - Line 288: Typo with "achieved" and "batch size".
  - Line 351: "chinchilla" should probably be capitalized.
  - Line 366: "They" -> "Then"?

---

> ### Author Response · Authors · 2024-11-15
> **Response [1]**
>
> Thank you for your constructive feedback. We have addressed each of your points as follows:
>
> > “EWA consistently improves model training efficiency”
>
> We included our code in Supplementary Materials. In the README.md or the assets folder, the experiments for cosine decay and WSD for various training lengths T. We can see that this is indeed the optimal training length for both of them. Note that these curves are at extremely high batch sizes, that’s why the decay indeed hurts as the noise induced by variance in the iterates is pretty small. We will still reword the statement to hold only in our setup, but we have done training length sweeps for cosine and WSD (for the 150m model), and we are comparing only after these sweeps.
> Note that we include the concrete sweeping settings in the Appendix. The decay phase is set to be last 0.1, 0.2, 0.3 of the training durations for all batch sizes. In the supplementary assets folder or README.md, we show that constant+EWA is competitive with Cosine and WSD with various decay ratios. fig. 9 (c) (d) (e) also shows that even if WSD reaches the target loss right at the end of training, it’s still comparable with constant+EWA.
>
> > The lines start to diverge a bit more for larger batch sizes, so is it possible that this is more pronounced for other settings (e.g. larger models, more training data, etc.)?
>
> Yes, this may be possible. However, based on Sec 2.3, we can expect that even if the curves diverge, that might happen beyond the critical batch size.
>
> > How the results and conclusions would change if one varies this parameter, e.g. to 10% or 50%
>
> Please note that it’s a design choice for choosing the overhead level depending on concrete training setups. The overhead level can be generalized according to specific practical settings and we found that these do not affect the fitted scaling laws too much. For example, when having the following 10%, 50% overhead, we get data scaling laws for 302M models of similar form as
>
> 10%: B* = 20.67 * D^(0.48)
>
> 20%: B* = 22.91 * D^(0.47)
>
> 50%: B* = 30.50 * D^(0.44)
>
> > width vs depth scaling.
>
> Please note that those settings are Chinchilla compute-optimal settings where the data size is scaled up proportionally with the model size. So large models are trained on more tokens. This means still increasing the scale via enlarging width or depth has similar CBS scaling behaviors, which we found in the next section to be mostly dependent on the data size.
>
> > In the legend of Figure 1b (left), what is the number in parenthesis? Is it the (relative) number of training samples?
>
> It is the token size, denoted by how many times or proportions the Chinchilla tokens are used. We revised the plot to improve clarity.
>
> > Plot size, font size, white space.
>
> Please refer to the current PDF. We have changed the formats of most plots substantially.
>
> > Linear scaling line as well as the 20% region and highlight the critical batch sizes, e.g. by a star marker?
>
> Please refer to current Fig. 5 for an illustration. The corresponding definition is on the left. Also, Note that connecting diagonal lines of squares in each plot can lead to the linear scaling regime.
>
> > Fig. 2 plot formats.
>
> Thank you for the reminder. We have updated the plot with new, more distinctive marker symbols to enhance visibility.
>
> > In line 110 you mention that "traditional learning rate decay strategies typically require predefining the total training duration" and you mention Defazio et al., (2024). However, isn't Schedule-Free a counter-example? Schedule-Free is also using a, at least to me, a relatively similar approach to your "constant+EWA" strategy, no (e.g. comparing the equation in line 158 to Eq. (5) by Defazio)? Could you elaborate on how your method differs from Schedule-Free?
>
> Sorry for the confusion. What we mean is how to get rid of fixed training duration is an actively studied problem so we cite Defazio et al., 2024. We have revised the narrative for clarity.
> Mathematically they are equivalent, overall ScheduleFree SGD can be understood as a accelarated SGD followed by weight averaging. This can be shown by denoting $y_t =(1-\beta) z_t+\beta x_t, z_{t+1}  =z_t-\gamma g\left(y_t\right), x_{t+1} =\left(1-c_{t+1}\right) x_t+c_{t+1} z_{t+1} $. Here, $y_t$ denotes the current weights of the model (note that the gradient is evaluated on $y_t$ ), while $x_t$ are the weights that will be used for evaluation. By recursively expanding $x_t$ to get that $x_t$ is an exponential average of $y_t$ for a constant $c_t$. However, ScheduleFree SGD couples the coefficients of momentum and weight averaging, which we decouple.

---

> ### Author Response · Authors · 2024-11-15
> **Response [2]**
>
> > For most of the paper, you sweep the batch size between  2^6  and  2^13 with the exception of Figure 3, where the batch size is between  2^16 and  2^22. Is this because in this Figure you count the number of tokens, so practically batch size * tokens per batch?
>
> Please note that as we explain in Sec 2.1, the batch size should be multiplied by context length to get the number of tokens per batch. So the range is still calibrated and we have made the x-axis to be **Batch Size (#Tokens)** instead of just **Batch Size**.
>
> > Did you try checking how close your thus predicted CBS tracks the true CBS for "unseen data points", i.e. test its predictive power?
>
> Due to computation constraints, we were not able verify the forecasted CBS.
>
> > In the concluding remarks you write "Our findings contribute insights into [...] particularly highlighting the role of hyper-parameters such as learning rate scheduling and optimizer settings" (line 530). What "optimizer settings" are you referring to?
>
> We are referring to Adam optimizers including $\beta_1, \beta_2$.
>
> > Nits and typos.
>
> Thank you for your careful read and we have fixed all the typos. We updated the draft significantly to address the formatting issues. Please double check the current versions.
>
> > Doesn't scaling from 151M to 302M double the depth, and therefore Figure 1 involves multiple ways of scaling up models?
>
> Please note that by default we use the scaling configurations in Tab. 2. So all 302M models except for Fig. 4 are scaled from 151M ones through enlarging depth instead of width.

---

> ### Comment · Reviewer_FgiN · 2024-11-21
> **Follow-up**
>
> Thanks for the detailed reply. I will respond point-by-point.
>
> **EWA improves model training efficiency**
>
> Maybe I am misunderstanding something here, but in the mentioned README.md (you mean the traj_2048.png, right?) you are not sweeping training lengths. For WSD you are showing three different lengths of the decay phase.
> My point is that WSD trains for 5000 steps while reaching the target before that (~4500 depending on the exact decay ratio). And your claim that EWA is better is based on "time-to-reach" a loss of 3.24. If I wanted to reach 3.24 as fast as possible with the WSD schedule, I would, e.g., try a training duration of 4500 steps (with 0.1 of the duration for the decay).
> My issue is that you are setting up WSD (or cosine) to train for T steps, but then compare the time to a result that happens before the T steps.
> An alternative comparison would be "I trained every schedule for 5000 steps, which schedule gives me the best performance after 5000 steps?" In your attached plot (traj_2048.png), it looks like constant+EWA is actually the worst here, with (at least) WSD (0.1) providing better performance.
> I don't mean to be too critical of the constant+EWA schedule, however, I don't think that the current evidence is enough to support a statement like "EWA consistently improves model training efficiency". I do believe that it is an interesting idea and getting rid of having to define a training duration is very relevant (I wonder, though, if it currently just trades one hyperparameter (training duration $T$) for another (EWA decay rate $\tau$). However, I believe the current claims are too bold for the provided evidence.
>
> **Diverging lines for CBS vs. context length**
>
> This might depend on the exact definition of CBS, no? E.g. if CBS uses 50% overhead, context length could affect the CBS. But I understand your point, though I still wonder whether larger models might show a different enough behavior such that the lines would diverge before the CBS (when using 20%).
> Should the reference in Section 2.3 be for Figure 4 (instead of Section 2.4)?
>
> **Effect of CBS definition**
>
> Thanks for the additional scaling laws. This is indeed interesting.
>
> **Width vs depth scaling**
>
> Thanks for the clarification. This was my mistake.
>
> **Plots**
>
> Thanks for revising the plots. The new legend in Figure 1 does indeed help.
> While I also appreciate Fig 5., my suggestion was more directed at adding the 20% overhead and highlighting CBS in all other plots. E.g., in Fig 1, it is hard for me to eyeball what exactly the CBS is. Perhaps adding a light gray shaded background region to indicate the 20% overhead could be helpful. And/or marking the CBS with a star instead of a circle for each line.
> Btw. I still think that some of the font sizes are quite small, e.g., the x-ticks in the bottom row of Fig 1. And the colored lines in Fig 2 have very similar and very bright colors, but the markers help.
>
> **Figure 1 only involves a single way of scaling up models.**
>
> I think I am misunderstand something here. In Figure 1, you are showing models between 85M and 1.2B parameters. According to Table 2, going from 151M to 302M doubles the number of layers (thus scales depth). Going from 151M to 604M doubles $d_{\text{model}}$ and the hidden size of the MLPs, while having the same number of layers. Isn't this width-scaling? So doesn't Fig 1 involve both width-scaling and depth-scaling models? I feel like I am misunderstanding the entire sentence perhaps?
>
> Thank you for addressing the other points as well. There, I don't have any follow-up questions. I am curious about the weight decay questions raised by Reviewer SERg and will be following the discussion.

---

> > ### Author Response · Authors · 2024-11-22
> > **Further Response and Results**
> >
> > Thank you for your detailed feedback.
> >
> > > Diverging lines for CBS vs. context length
> >
> > Please note that transformer-based LM model size is usually scaled up by enlarging MLP width or scaled in depth by increasing the number of transformer layers. These two are often independent of context length.
> >
> > Yes we are referring to Sec 2.3 and Figure 4.
> >
> > > Depth vs width.
> >
> > Yes your understanding is correct. Figure 1 involves both width and depth scaling. But the model size is not controlled (302M vs 604M). Instead Fig. 3 (right) is a controlled setting with both width and depth scaling.
> >
> > > Plot formats.
> >
> > Thank you and we will take into account your suggestions on plots. We found that adding the 20% line would overlap too much with existing curves and cause additional confusion.
> >
> > > Tuning of training durations.
> >
> > Please note that as indicated in Appendix, the baselines are well-tuned as we swept over learning rate [1e-3, 3.16e-3, 1e-2] for all schedulers and decay ratio [0.1, 0.2, 0.3] for WSD. For cosine scheduling, we first use optimal steps we found the relaunch that number of optimization steps as the maximum steps to make sure it would reach the the target loss right near the end of the training.
> >
> > > The impact of weight decay.
> >
> > We include the following experiments using decoupled weight decay for all the runs. We have hyper-parameter sweeps for 151M models using three schedulers - EWA, WSD, and ScheduleFree and our Constant+EWA: (due to time constraints we only swept over 4 batch sizes).
> >
> > lr - [3.16e-3, 1e-2], decoupled weight decay rate - [3e-04, 1e-04], batch size 512~4096, WSD decay ratio [0.1, 0.2, 0.3]. For the rest of hyper-parameters, we chose the best one from previous runs. We also sweep over the total training steps for both WSD and cosine to make sure they are close to optimal performance:
> >
> >
> > | Batch Size | Training Duration (steps)    |
> > |------------|-------------------------------|
> > | 512        | 11000, 13000, 16000          |
> > | 1024       | 8000, 8400, 10500                   |
> > | 2048       | 5800, 6400, 9500                   |
> > | 4096       | 3000, 3400, 3700             |
> >
> >
> > The results are in an anonymous link at https://docs.google.com/document/d/1edDHtur9019FJDqLJ0hILIZiKuV4YXsEaMf4gzTlkbg/edit?usp=sharing.
> >
> >
> > EWA+constant remains to be competitive. We see in the top figure that weight decay does slightly improve the schedule-free optimizer but not for others. All the schedulers remain nearly the same compared to the bottom figure reported in our paper. As our focus is the critical batch size, we additionally emphasize that adding weight decay does not (qualitatively) change the critical batch size too much.
> > If the reviewers believe the claim should be revised, we are happy to do so. For instance, we propose that “EWA consistently performs as a competitive choice compared to the other three schedulers we evaluated, within the specific settings of our study.”

---

> > > ### Author Response · Authors · 2024-11-25
> > >
> > > Thank you once again for your thoughtful review and valuable feedback! As we near the conclusion of the discussion period, we want to ensure that our previous responses have fully addressed all your concerns. If there are any remaining questions or unresolved points that we can clarify to help achieve a stronger evaluation, please don’t hesitate to reach out. We are more than happy to address this further.

---

> > > > ### Comment · Reviewer_FgiN · 2024-11-26
> > > >
> > > > **Figure 1 only involves a single way for scaling up models**
> > > >
> > > > I am still confused about this sentence regarding Figure 1 (but it is not crucial, and might just be my problem understanding the sentence). In your previous reply, you said "Figure 1 involves both width and depth scaling". What does the "single way for scaling up models" then mean? I thought it was referring to width vs. depth scaling. Or do you mean scaling up model size vs. scaling up data size?
> > > >
> > > > **Plot layout**
> > > >
> > > > Yes, I totally understand that my suggestions could make the plots more confusing and overloaded. The main challenge for me with the plots currently is that I can only estimate where the CBS is (e.g. when it exceeds 20% overhead).
> > > >
> > > > **Scheduler Comparison**
> > > >
> > > > I think the attached results demonstrate exactly what I meant. For example (looking at the first four plots, on page 1 of the linked document), now Cosine hits the target of 3.24 relatively close to the end of its schedule (e.g. shortly before the training duration). With this change, cosine is now faster for batch sizes 512, 1024, and 2048. For 4096 it is hard to see for me and I would say Cosine and Constant+EWA are about the same. So to me, the statement "EWA consistently improves model training efficiency. Training with EWA is slightly worse than Cosine learning rate decay for small batch sizes while outperforming Cosine for large batch sizes" is just not true.
> > > >
> > > > However, this doesn't mean there isn't any benefit to using Constant+EWA, precisely because no "training duration" has to be defined (or tuned). As such, a statement like "Without having to define training duration, EWA consistently performs similar to the other three schedulers we evaluated, particularly for large batch sizes, within the specific settings of our study. It thus is a viable choice for the remaining experiments in the paper." would be totally fine and correct.
> > > >
> > > > In the appendix, you mention "Longer training requires higher EWA decay rate $\tau$." Does this mean that $\tau$ needs to be tuned or chosen, similar to how the training duration would be in Cosine? I would appreciate a comment on how much easier EWA is in practice (or whether one hyperparameter is replaced with another).
> > > >
> > > > I hope you can understand why I am so stubborn about this point. I like the idea (and results) of Constant+EWA, but I believe the current statement is overclaiming.
> > > >
> > > > Assuming that you revise the claim about EWA, as you suggested, I have increased my score. Thank you for the productive rebuttal. I hope that my suggestions have helped to improve the paper.

---

> > > > > ### Author Response · Authors · 2024-11-26
> > > > >
> > > > > Thank you for your constructive feedback and raising the score. We appreciate your discussion which helped improve our paper.
> > > > >
> > > > > > In your previous reply, you said "Figure 1 involves both width and depth scaling". What does the "single way for scaling up models" then mean? I thought it was referring to width vs. depth scaling.
> > > > >
> > > > > Sorry about the confusion. As indicated in Tab. 2, scaling 151M to 302M involves doubling the depth but with width first. Scaling 151M to 604M involves doubling the width but with depth fixed. We are referring to these two different ways of scaling the model size. In contrast, as indicated by Tab. 5 and Fig. 3, there are two ways for scaling 151M to 604M: (1) doubling the width but with depth fixed (same as Fig.1); (2) making the depth 4x but with width fixed.
> > > > >
> > > > > > Scheduler
> > > > >
> > > > > Thank you for your suggestion regarding the statement; we will revise it accordingly. Please note that we didn't claim it's significantly better than the baselines in the main text. Also by "EWA consistently improves model training efficiency," we mean it in comparison to a constant learning rate without EWA. This observation is based on our experiments, where EWA demonstrated noticeable benefits in stabilizing training dynamics and achieving faster convergence across various setups.
> > > > > Yes, the decay rate needs to be tuned but as we can resume from checkpoints and each run with different decay rates are independent, we can launch them in parallel. But for cosine, we have sequential dependency, e.g. we need to decay schemes after observing the performance of previous runs and deciding the training duration each time.

---

### Meta-Review · Area_Chair_AsCi · 2024-12-23

**Metareview:**

The paper explores the scaling behavior of critical batch size (CBS) in autoregressive language model pretraining. CBS is shown to scale primarily with dataset size rather than model size, a finding supported by theoretical analysis. Empirical studies span models from 85M to 1.2B parameters, with controlled evaluations of depth and width scaling. The paper also assesses learning rate schedules, proposing a constant schedule with exponential weight averaging (EWA) as a competitive alternative.
Strengths highlighted by reviewers include the novelty of CBS scaling insights, clear experimental design, and the clarity of presentation through practical takeaways. Criticisms include the limited generalizability due to a limitation to the C4 dataset and relatively small model sizes. Incomplete hyperparameter tuning at larger scales also raised concerns about the validity of some conclusions. Despite these limitations, the paper offers valuable contributions to understanding CBS scaling.

**Additional Comments On Reviewer Discussion:**

Reviewer FgiN increased their score after EWA's benefits were clarified during the rebuttal, but still remained skeptical about overstated claims. Reviewer ivw4 increased their score based on improved clarity and better understanding of the relevance of results. Reviewer SERg raised the score after the authors included weight decay and better-tuned baselines and rephrased claims about EWA’s performance.

---

### Decision · Program_Chairs · 2025-01-22

Accept (Poster)